# International Migration Projections across Skill Levels in the Shared Socioeconomic Pathways



**Soheil Shayegh** [1,*] **, Johannes Emmerling** [1] **and Massimo Tavoni** [1,2]

1    RFF-CMCC European Institute on Economics and the Environment (EIEE), Centro Euro-Mediterraneo sui Cambiamenti Climatici, 20144 Milan, Italy; johannes.emmerling@eiee.org (J.E.); massimo.tavoni@eiee.org (M.T.)
2    Department of Management, Economics and Industrial Engineering, Politecnico di Milano, 20156 Milan, Italy
*    Correspondence: soheil.shayegh@eiee.org

**Abstract:** International migration is closely tied to demographic, socioeconomic, and environmental factors and their interaction with migration policies. Using a combination of a gravity econometric model and an overlapping generations model, we estimate the probability of bilateral migration among 160 countries in the period of 1960 to 2000 and use these findings to project international migration flows and their implications for income inequality within and between countries in the 21st century under five shared socioeconomic pathways (SSPs). Our results show that international migration increases welfare in developing countries, and closes the inequality gap both within and between low-skilled and high-skilled labor in these countries as well. In most developed countries, on the contrary, international migration increases the inequality gap and slightly reduces output. These changes are not uniform, and vary significantly across countries depending on their population growth and human capital development trajectories. Overall, while migration is strongly affected by inequality between developed and developing countries, it has an ambiguous impact on inequality within and between countries.

**Keywords:** migration; shared socioeconomic pathways; inequality; labor; demographics; human capital

## 1. Introduction

International migration fosters inter-cultural exchange and provides opportunities for people to relocate in search of better living conditions [1]. Despite a relatively stable trend over the last part of the twentieth century [2], the United Nations (UN) estimates that between 2000 and 2017 the stock of international migrants increased from 173 to 258 million people, accounting for about 3.4 percent of the global population [3]. This increase in international migration is expected to continue due to rising socioeconomic inequality and climate change, among other factors [4,5].

International migration is a complex decisionmaking process influenced by an array of social, economic, political, and environmental factors [6]. In the broader sense, these factors can be categorized into two groups, namely, pull factors and push factors [7]. Sociopolitical instability and lack of economic opportunities act as the main push factors, while the prospect of a better education, and higher economic and social stature have been identified as among the main pull factors attracting international migrants to host countries [8]. Therefore, a comprehensive study of international migration and its determinants requires analysis of the historical trends of the very large set of variables that may contribute to these migration decisionmaking processes at many different levels. The scope of this paper, however, is limited in that we aim to investigate only three main groups of factors that may have influenced migration flows over the past few decades: economic growth factors (GDP per capita), human development factors (population and education levels), and environmental factors (average temperature). Although there are many other factors that may have contributed to historical migration trends, we have chosen only those factors

with a contribution that has been validated by prior studies and for which the future trends are distinguishable among different socioeconomic pathways.

In this paper, we first examine whether and to what extent demographic change and income inequality have shaped historical international migration flows. We use our findings to show (a) how assumptions about future human capital accumulation and climate change will impact the projections of international migration flows, and (b) how the projections of international migration flows will change wealth distribution and income inequality within and between countries and skill groups by the end of the century.

Global studies of migration have been largely divided into two camps. One group has used statistical models to identify the key determinants and drivers of international migration and then used these models to project future trends [9,10]. We follow the leading literature in this category to develop a broad gravity model of bilateral migration flows among 160 countries for the period of 1960–2000, allowing us to identify the key drivers of international migration. However, relying solely on econometric models for projecting the future outcomes requires strong limiting assumptions about migration trends. For example, UN projections of international migration have been largely based on a simple assumption that levels of net migration will remain constant at their current levels until 2045–2050 and then will gradually decline to 50 per cent of the projected level of 2045–2050 by 2100 [11,12].

In order to improve these insights using statistical methods, another group of researchers attempt to account for the interdependencies in future distributions of income, population, and migration by developing structural models of economic growth [13,14]. These models provide an underlying explanation for migration mechanisms that can be used to assess future migration projections under different socioeconomic scenarios [15,16]. In a similar fashion, in this paper we develop a structural model to project future migration flows under five distinct shared socioeconomic pathways (SSP) scenarios [17]. This model allows us to observe the evolution of migration in the broader context of future socioeconomic changes and to examine the interaction between migration, demography, human capital, income inequality, and economic growth within a unified framework [15,18]. Therefore, the first contribution of our paper is to develop a structural model that shows how underlying human capital accumulation trends within various SSP scenarios will shape the international migration patterns throughout the 21st century from the perspective of both sending and receiving countries.

The broader impact of migration on economic development and inequality has been long debated in the literature. While several previous studies show a negative impact of migration on wages in receiving regions [19], this impact does not appear to be homogeneous across skill levels. Immigration has been shown to have a negative impact on wages in low-skilled labor and a positive impact on wages in high-skilled labor [20,21]. Other studies have both ruled out a negative impact of immigration on the wages of native labor in receiving regions [22] and shown a positive increase in the GDP of receiving countries [23]. However, except for a few global studies [13,15], most of the literature in this field has been based on historical data from specific countries or regions. Hence, the second contribution of our paper is to identify the impact of international migration on income inequality and social welfare in sending and receiving countries across different SSP scenarios.

Finally, in addition to socioeconomic factors, environmental parameters can play a significant role in shaping migration trends. Recent studies on the impact of climate change on demographic change and population dynamics have highlighted important mechanisms through which climate change alters human capital accumulation and may trigger international migration as an adaptation response to extreme events [24–27]. These studies are mainly focused on historical migration trends, and try to demonstrate a positive and statistically significant relationship between environmental factors (e.g., temperature and precipitation) and international migration [28–30]. While several studies have found no evidence of impact of climate change and natural hazards on international migration [31], other studies have highlighted the indirect effects of environmental factors on migration through changing wages [32], especially in agricultural economies [33]. Our statistical

analysis demonstrates no evidence of a direct impact of temperature rise on international migration flows. Nevertheless, we include an explicit form of climate damage function in our structural economic model to account for future economic losses due to climate change and their impact on migration flows. In this manner, our paper is close to several of the most recent works on the intersection of international migration, inequality, and climate change [18,34,35]. It is worth mentioning that the emphasis of our model is on the long-term gradual movement of people over 20-year time periods, and it therefore does not account for the impacts of natural disasters and climate shocks on migration flows (i.e., crisis-induced migration [36]). Instead, we integrate representative concentration pathways (RCP) into our SSP scenario framework that allow us to consider a full spectrum of socioeconomic and climate factors concurrently. The third contribution of our paper therefore lies in its ability to combine statistical and structural models with SSP and RCP scenarios in order to develop migration projections that can be interpreted within future environmental and socioeconomic contexts.

## 2. The SSP–RCP Scenario Framework

Figure 1 shows the outline and components of our modeling framework.

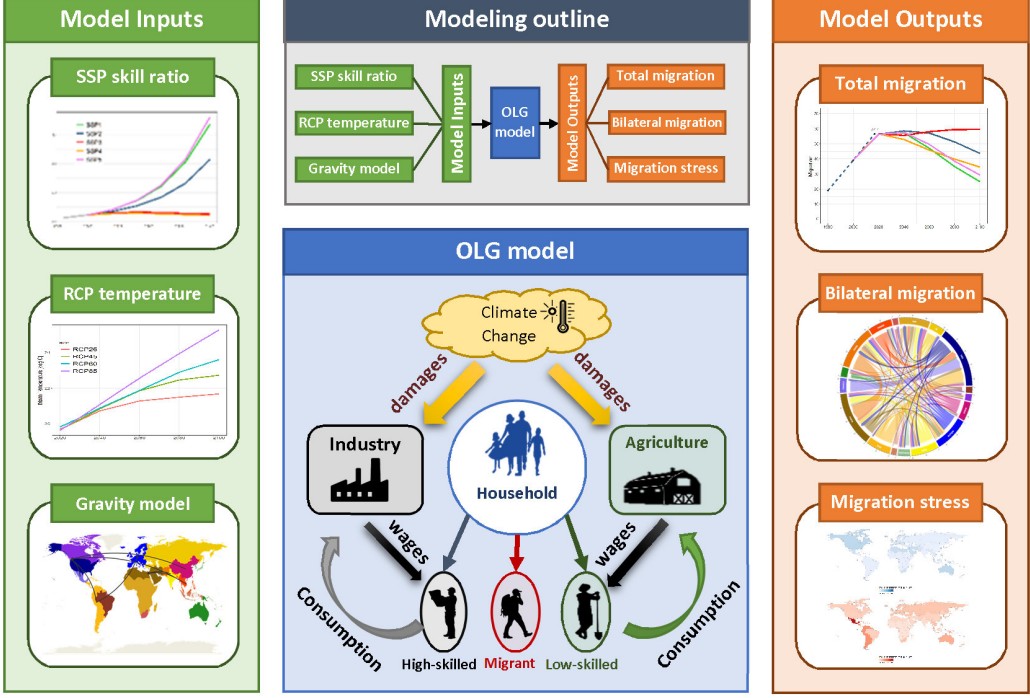

**Figure 1.** Modeling outline and its components. An SSP database is used to generate projections of skill ratio. RCP scenarios provide temperature projections. The historical bilateral migration data are used to generate a gravity model of migration (Figure 2). The OLG model is calibrated with these inputs and generates total migration flows (Figure 3), bilateral migration flows for each SSP scenario (Figure 4), and migration stress maps.

The first inputs to our model were the SSP projections of human capital development for each country. These are quantified in our model as a skill ratio (e.g., the ratio of high-skilled adult population to low-skilled adult population) across SSP scenarios. Each SSP scenario has a unique trajectory of key socioeconomic indicators (see Figure A2 in Appendix C.6, which is based on data from [17,37,38]). Under each SSP scenario, countries experience different rates of economic growth and human capital accumulation trajectories [38]. As a result, socioeconomic conditions under each SSP may act as pull or push factors that will drive international migration flows in a non-uniform and heterogeneous

way all across the world [39]. The underlying narratives of the SSP scenarios are presented in Appendix A.

In addition to the SSP projections of human capital accumulation, explicit assumptions about future climate change were integrated into our model by including two RCP projections of country-specific temperatures. We report the main results of our model under RCP6.0 (*baseline case*), where global temperature increases by about 3 °C above pre-industrial levels by the end of the century [40]. The results associated with a lower climate change scenario, RCP2.6 are reported in Appendix D.3.

Finally, we developed a gravity model of historical bilateral migration data for 160 countries to find the key drivers of international migration. We found that the probability of future migration (i.e., the percentage of migrants to the total population) depends on current population, income inequality, and the distance between the sending and receiving countries. We used the coefficient of the gravity model to project the future migration trends within an overlapping generations (OLG) model of population dynamics.

At the heart of our approach is the OLG model of population dynamics, which is based on the SSP-RCP scenario projections and the results of the gravity model of bilateral migration probabilities. The country-level results of the OLG model projections are aggregated and presented at the regional level by dividing the world into seventeen regions. Each region includes countries with similar socioeconomic status and geographical proximity; see Figure 2 and Table 1 for the definitions of the regions.

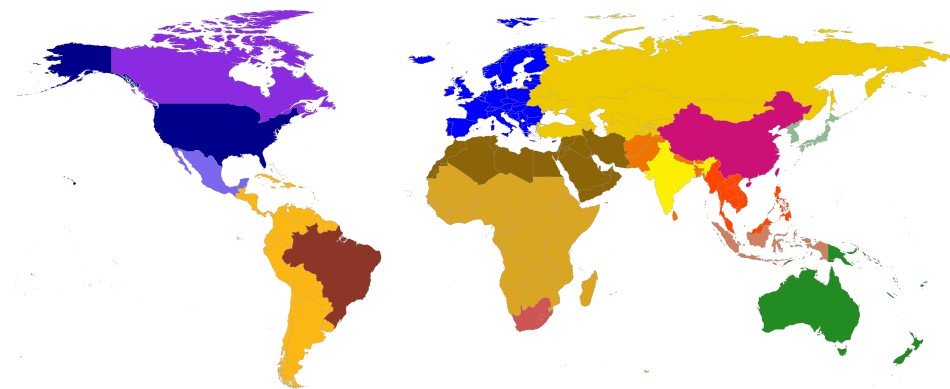

**Figure 2.** Seventeen geographical regions used for calculating global migration flows.

**Table 1.** The seventeen geographical regions used to calculate global migration flows along with the color code used in Figures 2 and 4.

| Region | Abbreviation | Color |
|---|---|---|
| Canada | CAN | |
| Japan-Korea | JPN | |
| Oceania | OCE | |
| Indonesia | IDN | |
| South Africa | ZAF | |
| Brazil | BRA | |
| Mexico | MEX | |
| China | CHN | |
| India | IND | |
| Non-EU Eastern European | TEC | |
| Sub Saharan Africa | SSA | |
| Latin America-Caribbean | LAC | |
| South Asia | SAS | |
| South East Asia | SEA | |
| Middle East-North Africa | MEA | |
| Europe | EUR | |
| USA | USA | |

## 3. Econometric Model

Estimating future international migration patterns requires analysing bilateral migration patterns and behaviors in the past. The historical accounting of migration patterns, however, is a challenging task due to the lack of data, inconsistencies in measuring and reporting data, and other discrepancies [41]. Nevertheless, estimating historical bilateral migration flows is a relatively reliable approach for establishing historical patterns of migration between countries [42]. This can be done using (a) stock differences, (b) migration rates, or (c) demographic accounting [43]. In this paper, we used stock differencing (calculating the difference in the size of migrant stocks at the beginning and end of a given time period for a given pair of countries) to calculate the historical bilateral migration flows between 160 countries from 1960 to 2000.

To account for the determinants of bilateral migrations, we included well-established factors related to characteristics of the origin and destination, including economic conditions, population, and skill levels. Due to the lack of data and inconsistencies in measuring and comparing immigration policies [44], we did not model migration policies explicitly. However, these factors and other social and cultural factors are implicitly accounted for in our econometric model through the inclusion of country-specific fixed effects.

The main feature of our econometric model is the separation of migration probabilities across skill levels. The impact of education and skill level on migration probability has been widely observed, and notably, a skill bias has been frequently found or discussed in migration patterns [45]. In order to calibrate the bilateral migration probabilities in our model, we used data on population and educational attainment [38] which include projections until 2100 for all the SSP scenarios. Moreover, we used historical data on GDP, income inequality, and population from the IMF's World Economic Outlook [46,47]. Historical bilateral migration flow data were obtained from decadal data on country by country migration stocks [48]. Based on these country-level data we estimated a gravity model equation of migration across countries similar to previous empirical studies on the determinants of migration. In particular, we used the following equation:

$$\frac{M^k_{ij,t+1}}{L^k_{i,t+1}} = \theta^k_1 log(L^k_{i,t}) + \theta^k_2 log(L^k_{j,t}) + \theta^k_3 log\left(\frac{GDPPC^k_{j,t}}{GDPPC^k_{i,t}}\right) + \theta^k_4 log(d_{ij}) + \eta_i + \delta_j + \varepsilon_{ij} \quad (1)$$

That is, at every time period $t$ we estimated the probability of migration of adults of skill level $k$ from sending country $i$ to receiving country $j$ at the next time period $t+1$. This probability is calculated as the future flow of migrants ($M^k_{ij,t+1}$) per total adult population of the sending country ($L^k_{i,t+1}$). As in our economic model, the migration probability is a function of the current adult population in origin ($L^k_{i,t}$) and in destination ($L^k_{j,t}$), the wage ratio of receiving to sending country, and the distance between both countries ($d_{ij}$). We used the per capita output of adult groups of skill level $k$ measured in USD-PPP of 2005 ($GDPPC^k_{j,t}/GDPPC^k_{i,t}$) as a proxy for the wage ratio of the receiving to the sending country. We computed this by dividing the adult population into high-skilled and low-skilled groups based on education attainment records. Using historical Gini index data for each country, we computed the wage for each skill group by matching the Gini index and per capita GDP (two equations and two unknown wages). The distance was taken as the shortest distance between the borders of each two countries. Note that because the formula needs to be implemented in the analytical model, we used the simple linear probability model (LPM) to estimate this equation and include both origin ($\eta_i$) and destination ($\delta_j$) fixed effects to account for country-specific idiosyncratic drivers of migration.

Alternatively, Equation (1) can be estimated using the Pseudo-Poisson Maximum Likelihood (PPML) estimator from [49] to account for zeros in the observed migration probabilities and heteroskedasticity (see Appendix D.1 for a discussion of the suitability of the PPML model in projecting future migration probabilities). However, we found that in terms of the relevant drivers, the estimated coefficients are rather similar.

Equation (1) can be estimated using different specifications. For model calibration, our central estimation was a standard OLS estimation of the migration probability such that the correlation with GDP, population, and distance are taken into account in the model. We included country-specific autonomous terms. The results of the econometric model are presented in Appendix B.

## 4. Overlapping Generations Model

We developed a structural model of population dynamics and economic growth to project future global population growth and international migration flows. In this model, we accounted for two groups, high-skilled and low-skilled labor, working in the agricultural and non-agricultural sectors, respectively. Individuals live through two periods, childhood and adulthood. Decisions about the number of children (population growth) and their skill level (human capital) are made by parents through the optimization of the parental utility function. Parents are altruistic, which means their utility increases through the expected well-being of their children as well as through consumption of goods. There is a quantity–quality trade-off in raising children, in the sense that high-skilled children with potentially higher income in their adulthood require more parental investment in their childhood. We calibrated this model to reflect the projection of human capital development for each country under the five SSP scenarios. In each period, the projections of climate conditions for the next period were obtained from the RCP projections and used to calculate climate-induced damages in each economic sector. In addition, the probability of bilateral migration of high-skilled and low-skilled adults is endogenous, and was calculated based on the results of the gravity model in Equation (1) in Section 3. We modified the migration probability specified in this equation as follows in order to use it in our OLG model:

$$\beta_{ij,t+1}^k = \theta_{ij}^k + \theta_1^k log(L_{i,t}^k) + \theta_2^k log(L_{j,t}^k) + \theta_3^k log\left(\frac{w_{j,t}^k}{w_{i,t}^k}\right) + \theta_4^k log(d_{ij}) + \eta_i + \delta_j, \quad (2)$$

where migration probability ($\beta_{ij,t+1}^k$) indicates the portion of individuals with skill level $k$ from country $i$ that will migrate to country $j$ at the next time step $t+1$. This probability is a function of the current population of adults with the same skill level $k$ in the sending country ($L_{i,t}^k$) and receiving country ($L_{j,t}^k$), the current ratio of wages between the same type of labor in two countries ($\frac{w_{j,t}^k}{w_{i,t}^k}$), the distance between the two countries ($d_{ij}$), and the sending and receiving countries' fixed effects ($\eta_i$ and $\delta_j$). The parameter $\theta_{ij}^k$ is the migration residual between the two countries, calculated by fitting the probability equation to historical migration data from 1980 to 2000. In a modified version of Equation (A1) in Appendix D.2, we have implemented an exogenous policy parameter to this equation to account for migration policy variations across SSP scenarios. Detailed descriptions of the OLG model, its calibration, and the solution methodology are provided in Appendix C.

## 5. Projections of Migration and Demographic Outcomes

The output of the OLG model is a set of future migration flows across the SSP scenarios, as shown in Figure 1. As outlined in Section 2, each SSP revolves around a unique narrative about future socioeconomic and technological developments. For example, while SSP1 and SSP5 make very different assumptions about fossil-fuel consumption and related emissions, they have very similar population growth and human capital accumulation trajectories [38]. As a result, the projections of total migration flows overlap in these two scenarios. As our gravity model shows, and as the comparison between the results of our *baseline case* here and the RCP2.6 case in in Appendix D.3 further suggests, we can safely assume that gradual climate change (as opposed to climate shocks and catastrophes) does not have a direct impact on long-term migration trajectories. Although the SSP5 narrative assumes higher levels of international migration compared to SSP1, our baseline OLG model, which uses only SSP skill ratio as an input, is not able to capture such differences without further

assumptions about migration policies. Therefore, in Appendix D.2 we have introduced a modified version of Equation (A1) with an exogenous SSP-specific migration policy parameter in order to allow the model to distinguish between the migration dynamics in the SSP1 and SSP5 scenarios.

Another important driver of migration is the income inequality between receiving and sending regions, represented by the wage differences between countries in Equation (A1). The impact of income inequality on migration flows can be observed by comparing the SSP2 and SSP4 scenarios. These two scenarios have similar global population growth along with very different human capital accumulation trajectories [38]. The SSP2 scenario projects moderate growth in human capital accumulation, especially in developing regions. The SSP4 scenario, on the other hand, projects a halt or even a decline in human capital growth for most regions. As a result, SSP4 experiences higher inequality and much larger migration flows. Throughout our analysis, we refer to the difference between the results of these two scenarios as the *inequality effect*.

Finally, climate change-induced damages in different sectors can potentially create new mechanisms for labor movement within and between countries. Here, we present the results of our analysis for the case of RCP6.0 temperature trajectories (i.e., the *baseline case*). The results for a case with limited climate change (RCP2.6) are presented in the in Appendix D.3 and show a very similar pattern to the results of the *baseline case*.

Panel (a) in Figure 3 shows the total migration flows for every 20 years under the five SSP scenarios. In the SSP1, SSP2, and SSP5 scenarios, the migration flow peaks around 2060 and declines afterwards. In the SSP3 and SSP4 scenarios, however, the migration flow rapidly increases until it reaches its maximum by the end of the century. Comparing the results under SSP2 and SSP4 reveals that the *inequality effect* changes both the size and shape of the migration flows. We can further distinguish the migration flows by the skill level of migrants. Panels (b) and (c) in Figure 3 show projections of the global flow of high-skilled labor and low-skilled labor, respectively. In the SSP1, SSP2, and SSP5 scenarios, high-skilled migration accounts for the majority of total migration flows while in SSP3 and SSP4 it accounts for about half of total migration.

Figure 4 shows the bilateral flow of migrants across seventeen aggregated regions in the period of 2080–2100 for the five SSP scenarios and compares it with historical bilateral flows from the 1980–2000 period. The circular plots show the total number of migrants on a circle with arrows pointing to and from each of the seventeen macroregions our country-level data were aggregated to. The numerical scale on a circle indicates the number of total migrants in hundred thousands. Regardless of underlying SSP assumptions, there are several notable migration trends in all the SSP scenarios (e.g., Europe emerges as a main destination while India becomes a major sending region). The SSP1 and SSP5 graphs in Figure 4 show very similar composition of bilateral flows. In contrast, comparing the SSP2 and SSP4 graphs paints a different picture; while the historical bilateral migration flows are dominated by a few main trends (e.g., Mexico and Latin America to the U.S.), the medium range growth rate of economy and education under the SSP2 scenario leads to a more homogeneous future flow of migrants between developing and developed regions. In the inequality-driven world of the SSP4 scenario, however, migration trends remain dominated by South-to-North or Developing-to-Developed flows, notably from India and Sub-Saharan Africa to Europe.

Inequality in skill and education both drives migration and evolves as a result of migration and the subsequent redistribution of wealth and labor in the sending and receiving countries. In our dynamic model of population dynamics and migration, income disparity between the two types of labor and the heterogeneity of climate change damages creates incentives for parents to have a specific number and type of children in each period. This in turn changes the migration probabilities and induces movement of labor among countries. Once all bilateral migrations are completed, labor markets are cleared and new wages realized, and income inequality within and between regions evolves accordingly. Figure 5 shows a snapshot of the inequality measures for the year 2080 across the SSP scenarios for

the two cases, namely, with migration and without migration. In the case without migration, we assume zero bilateral migration; therefore, the inequality indicators are driven only by demographic change and labor allocation within each country. In order to quantitatively compare the results, we focus on the SSP2 scenario as a middle-of-the-road scenario.

Panel (a) in Figure 5 shows the relative wages of low-skilled labor in each region compared to the global average. In developed regions (depicted in darker colors, starting with the U.S. at the top and moving clockwise), the wages of low-skilled labor are much higher than the average in the case without migration. For example, in Canada (CAN) the wages of low-skilled labor are about 3.8 times the global average in the case without migration; however, when migration is allowed they are much lower, at about 0.2 times the global average. Meanwhile, the wages of low-skilled labor in developing countries improve to close the global inequality gap for low-skilled labor when migration is allowed. For example, the wages of low-skilled labor in South East Asia (SEA) increase from 0.03 times the global average in the case without migration to about 0.2 times the global average in the case with migration. Overall, migration reduces global inequality among low-skilled labor across the SSP scenarios, while the rate of reduction is lower in SSP3 and SSP4.

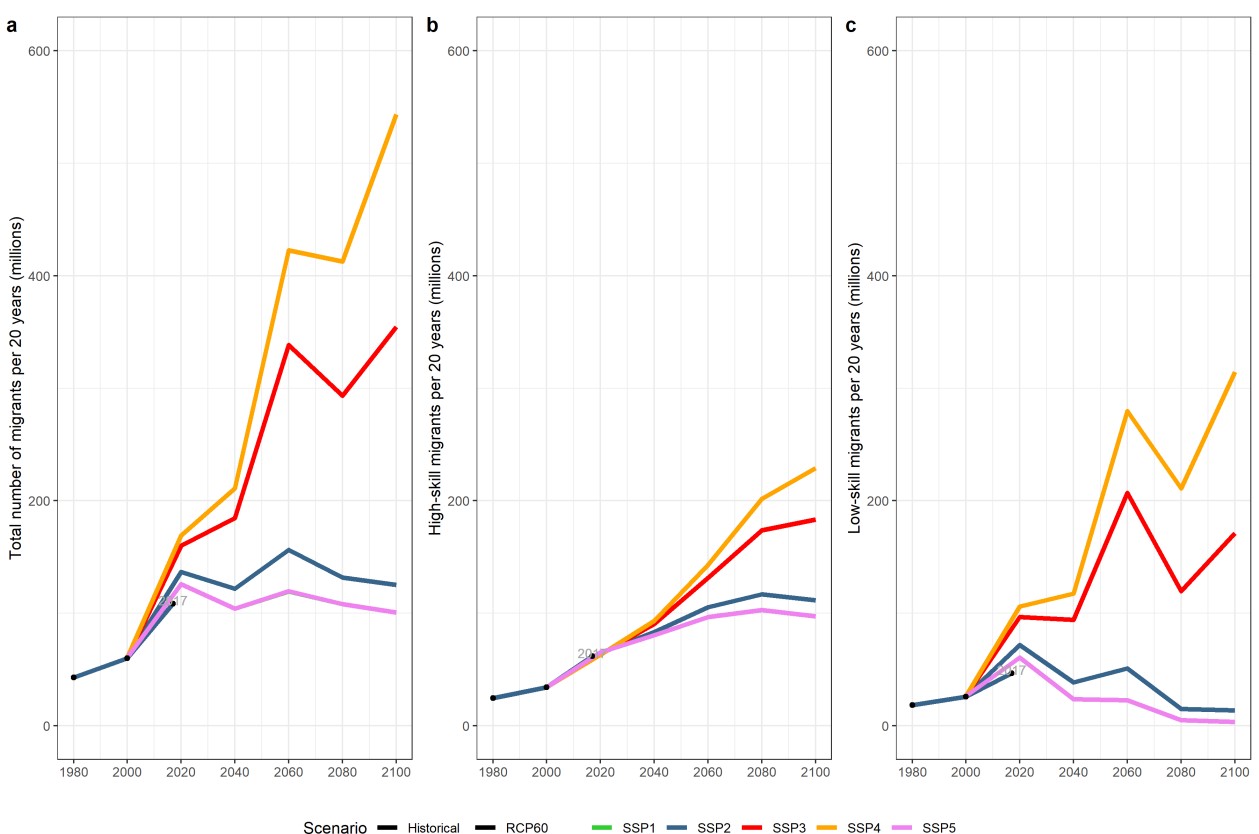

**Figure 3.** Historical and projected twenty-year global migration flows across the SP scenarios: (**a**) Total global migration; (**b**) High-skilled global migration; (**c**) Low-skilled global migration. The global share of skilled migrants is around 57% of total migrants, compatible with existing estimates of the skill-bias of migration. The focus on periods of twenty years reflects the duration of a generation in our OLG model. We have aggregated historical migration data for each decade to obtain the twenty-year historical flows. Historical data for the 2000–2010 period are excluded due to their inconsistency.

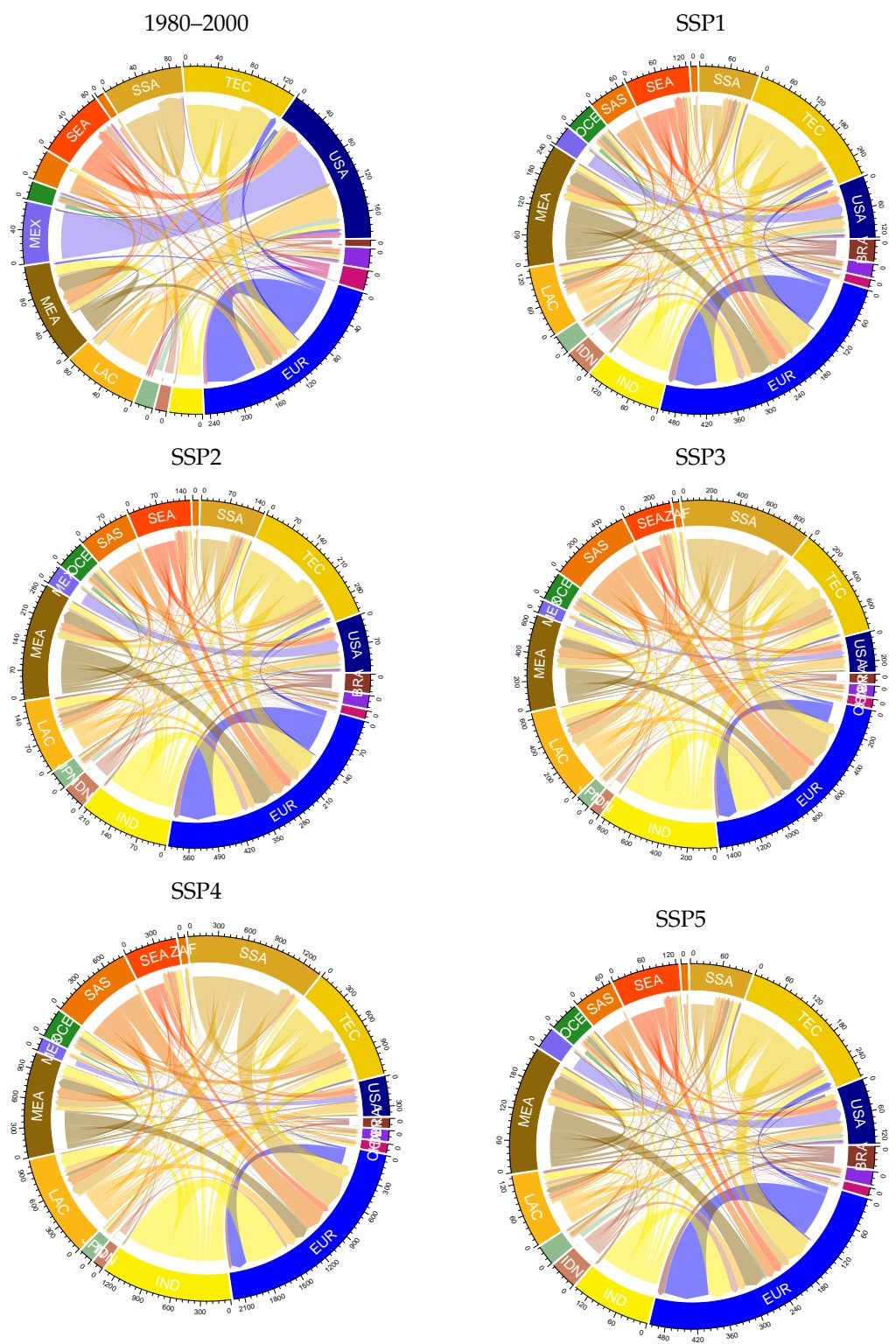

**Figure 4.** Regional bilateral migration flows in the SSP scenarios for the 2080–2100 period, annually in 100 thousands.

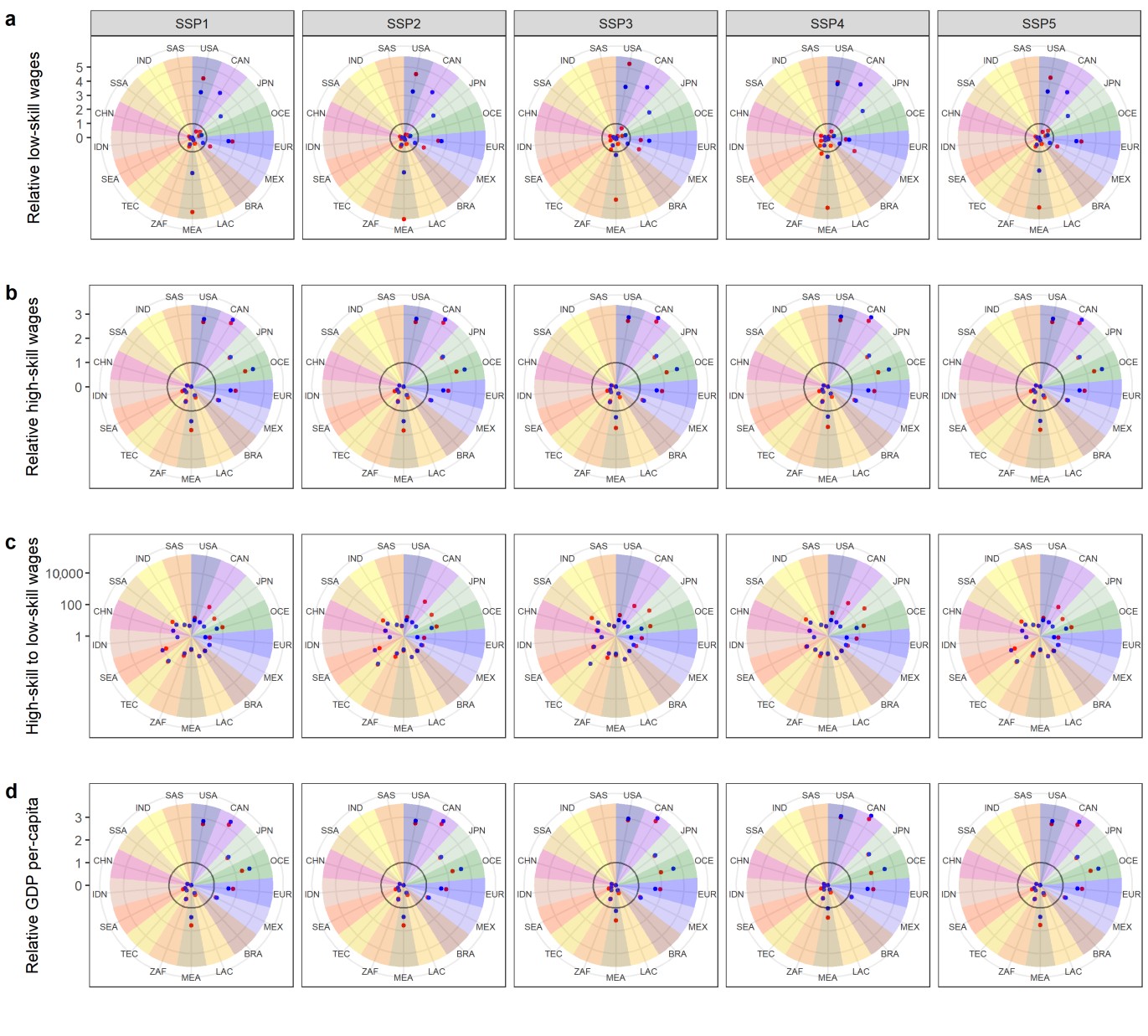

**Figure 5.** Income inequality by regions in 2080 in the SSP scenarios with and without Migration: (**a**) Relative wages of low-skilled labor to global average; (**b**) Relative wages of high-skilled labor to global average; (**c**) Relative wages of high-skilled labor to low-skilled labor in logarithmic scale; (**d**) GDP per capita relative to global average. The dark circle in the middle in rows (**a**,**b**), and (**d**) identifies the global average. The regions are ordered based on per capita GDP in the year 2000, starting from the USA with the highest and moving clockwise to the Sub-Saharan Africa (SSA) with the lowest.

Panel (b) in Figure 5 shows the relative wages of high-skilled labor in each region compared to the global average. The overall change in the relative wages of high-skilled labor seems to follow a similar pattern, though at a lower rate. For example, in Canada the wages of high-skilled labor are about 3.3 times the global average without migration, and they fall slightly below this level when migration is allowed (3.1 times the global average). On the other hand, the wages of high-skilled labor in developing countries slightly increase under most of the SSP scenarios. For example, the wages of high-skilled labor in South Africa (ZAF) remain around 0.6 times the global average with or without migration. Overall, global inequality among high-skilled labor is slightly improved by migration.

In order to compare the inequality within each region, we calculate the wage ratio of high-skilled labor to low-skilled labor in panel (c) in Figure 5. In developed regions, the inequality between high-skilled and low-skilled labor increases with migration. This increase is even higher in the inequality-driven SSP4. For example, in the US the wage ratio of high-skilled to low-skilled labor increases from 10.6 in the case without migration to 16 in the case with migration. At the same time, income inequality in developing regions slightly declines or remains unchanged. For example, in Latin America (LAC) the wage ratio of high-skilled to low-skilled labor decreases from 28 in the case without migration to 27 in the case with migration. This is in line with studies showing a positive impact of migration in reducing inequality, especially in sending regions [50]. Overall, we observe that while migration reduces global inequality between regions, especially for low-skilled labor, it increases inequality within developed regions. This is in part because of the reduction in the wages of low-skilled labor in receiving regions due to immigration.

Finally, panel (d) in Figure 5 demonstrates global inequality in terms of output per capita. The results are to a certain degree similar to the pattern observed in panel (b) for the relative wages of high-skilled labor. While migration slightly decreases the relative GDP per capita in developed regions, it increases the relative GDP per capita in developing regions.

## 6. Discussion

In this paper, we developed a modeling framework for projecting bilateral international migration flows throughout the 21st century across five different SSP scenarios. We used historical bilateral migration data for 160 countries in a gravity model in order to take into account both natural drivers (e.g., distance) and socioeconomic drivers (e.g., wage disparity between receiving and sending regions). We applied our empirical findings to an overlapping generations model [34,35] to generate projections of population growth, demographic change, and migration flows for each country. Our findings can be summarized in four categories. Previous studies of the international migration projections across SSP scenarios have shown the role of sociodemographic factors in shaping future migration patterns [51]. In our study, we show the additional role of income inequality in driving future migration flows, and highlight the fact that wage disparities among labor in receiving countries create incentives in the migration of high-skilled labor in scenarios with higher human development prospects (SSP1, SSP2, and SSP5). In contrast, the flow of low-skilled migrants is higher in scenarios with lower human development trajectories (SSP3 and SSP4). Therefore, the first contribution of our study is to underline the role of socioeconomic assumptions, including inequality within and between countries and skill groups, in shaping future migration patterns. For example, while SSP2 and SSP4 share similar population growth trajectories, due to their different human capital development trajectories (e.g., different skill ratios) the migration scale and pattern in these two scenarios diverge greatly.

Furthermore, while recent studies have shown an overall positive impact of migration on global wealth creation across all SSPs [52], our results indicate a much more complex picture where such impact is not homogeneous. Therefore, the second contribution of our study is to show the heterogeneous impact of migration on inequality across countries, skill levels, and SSP scenarios. For example, we show that while international migration has a clear impact on closing the inequality gap among low-skilled labor, its impact on closing the inequality gap within high-skilled labor is less obvious. On the other hand, we find that the inequality gap between high-skilled and low-skilled labor increases significantly in developed countries while being reduced slightly in developing countries as a result of international migration. In other words, while developing countries experience higher inequality at the beginning, migration eventually exerts greater pressure on equality in receiving countries that are predominantly developed.

The final contribution of our study is to highlight the economic benefits of migration. By comparing migration flows over time and across SSP scenarios, we observe that while major developed regions such as Europe gain positive economic benefits from migration at

the cost of increasing inequality, other developed countries experience both an economic loss and an increase in inequality as a result of migration. On the other hand, most developing regions gain economic benefits from migration, and manage to close the inequality gap both within and between their low-skilled and high-skilled labor forces.

## 7. Conclusions

Our research sheds light on the importance of the interconnecting links between three major factors in socioeconomic development: mobility, inequality, and education. We have shown that human mobility and migration at the global scale is mainly driven by sociodemographic factors, including income inequality between the sending and receiving countries.

International migration, on the other hand, can change the wage dynamics in both sending and receiving countries, creating demand for certain types of labor and as a result changing the overall wealth created in a society. We show that while certain developed countries might benefit from receiving high-skilled migrants from developing countries, the key winners in the international migration game are the sending countries, which benefit from increased demand for higher education and higher wages in the future.

Although we used the most readily-available set of bilateral migration data for 160 countries over 40 years to calibrate our econometric model, the results of our study must be interpreted with caution. First, our model does not explicitly consider bilateral migration policies, which play a key role in restricting or facilitating migration flows [53]. Second, environmental factors such as climate change were applied in our OLG model through a calibrated damage function. While this model is thus capable of reflecting economic losses due to gradual increase in global mean temperatures, it does not account for other climate change factors such as sea-level rise [18]. Finally, due to its reliance on long-term economic optimization of households' well-being in equilibrium, our model is not capable of accounting for fast-onset shocks such as wars and civil conflicts, which can create forced migration flows in the short term. As a remedy to these shortcomings, future research in bilateral migration can build on our study by expanding its econometric model and including other traditional and nontraditional sources of migration data [41]. Future research can additionally expand and further enrich the structural model used in this study (i.e., the OLG model) by considering other sources of income and expenses, such as remittances and retirement savings. The economic sectors can potentially be expanded to include the service sector and the skill composition of labor in this sector.

Despite the aforementioned limitations, our study contributes to ongoing debates on international migration and economic development in three clear ways: (1) by highlighting the impact of inequality in driving migration, especially among low-skilled labor, we draw the attention of policymakers in developed countries to the role of global equity in shaping future migration flows to their countries; (2) we show that while facilitating migration can ease inequality pressures in sending countries, if not managed properly it can widen the inequality gap in the receiving countries; (3) we underline the heterogeneous impact of migration on inequality in different countries and among different labor groups. We have shown that when it comes to international migration, there is no one-size-fits-all policy that can benefit all players. Therefore, it is important that future migration policies be co-developed in close collaboration with all stakeholders and potential beneficiaries.

**Author Contributions:** Conceptualization, S.S., J.E. and M.T.; methodology, S.S. and J.E.; validation, S.S. and J.E.; formal analysis, S.S. and J.E.; investigation, S.S. and J.E.; resources, S.S. and J.E.; data curation, J.E.; writing—original draft preparation, S.S. and J.E.; writing—review and editing, S.S., J.E. and M.T.; visualization, S.S. and J.E.; supervision, M.T.; funding acquisition, J.E. All authors have read and agreed to the published version of the manuscript.

**Funding:** This research was funded by the European Union's Horizon 2020 research and innovation programme under grant agreements No. 821124—NAVIGATE—Next generation of AdVanced InteGrated Assessment modelling to support climaTE policy making.

**Institutional Review Board Statement:** Not applicable.

**Informed Consent Statement:** Not applicable.

**Data Availability Statement:** Data obtained from the World Bank and publicly available at https://databank.worldbank.org/source/global-bilateral-migration (accessed on 12 April 2022).

**Conflicts of Interest:** The authors declare no conflict of interest.

## Appendix A. SSP Narratives

**SSP1** is a sustainability scenario with a medium level of migration where the world gradually moves toward low-emission technologies with higher rates of education and human capital accumulation. Investments in education and health accelerate the demographic transition, and inequality is reduced both across and within countries.

**SSP2** is a middle-of-the-road scenario with a medium level of migration and moderate challenges to mitigation and adaptation. In this scenario, the world follows a business-as-usual development path. It experiences environmental degradation with a decline in intensity of resource and energy use. Global population grows moderately and levels off in the second half of the century. There are slow improvements in income inequality and environmental challenges.

**SSP3** represents a world filled with regional rivalry, with a low level of migration and high challenges to mitigation and adaptation. In this world, nationalism and concerns about competitiveness and security provoke regional conflicts and a decline in investments in education and technological development. Slow economic development coupled with high population growth in developing countries exerts pressure on natural resources and environmental indicators.

**SSP4** is an inequality scenario with medium levels of migration where economies are growing in different rates across the globe. The world is divided between highly educated societies with access to high technologies, capital, and resources, and lower-income poorly educated societies with low-tech economies.

**SSP5** represents a world with a high level of migration where economic development is closely tied to high fossil-fueled consumption. Technological progress is rapid and human capital grows fast. Investments in health, education, and institutions increase as global population peaks and then declines into the 21st century.

## Appendix B. Econometric Model: Data and Results

Table A1 summarizes the data used for the gravity model estimation. The main data sources are the historical population and education data based on [38]. Based on the education and age group data, we compute the adult population (15–70 years) according to different educational attainment and the population of children (those less than 15 years old). The GDP data are based on combining IMF-WEO historical data and the SSP projections from [47]. In order to compute the wage ratio and GDP per capita for each of the two skill-level groups, we combine the GDP data with inequality data based on income quintiles from the World Development Indicators.

Table A2 shows the estimation results for the whole population and separated estimations for high-skilled and low-skilled adults. The results confirm a strong effect of relative wages between origin and destination region, especially for low-skilled labor. Moreover, the population of the origin country has a positive effect, with a higher coefficient for low-skilled migrants. Distance has a negative and robust impact on migration, as expected.

Although we use bilateral migration data from 1960 to 2000 in our econometric model, we only show the 1980–2000 data in the forthcoming graphs for the purpose of comparison with the SSP projections throughout the paper. Furthermore, we use the 1980–2000 data to calibrate and match the migration probabilities calculated in the structural model in the next section with the 2000 migration data.

**Table A1.** Summary Statistics.

| Statistic | N | Mean | St. Dev. | Min | Max |
|---|---|---|---|---|---|
| population_origin | 38,416 | 35.929 | 127.241 | 0.115 | 1269.117 |
| governance_origin | 24,800 | 0.531 | 0.195 | 0.128 | 0.954 |
| gdppc_origin | 38,416 | 9862.675 | 15,351.970 | 176.507 | 105,723.400 |
| gini_origin | 27,696 | 0.390 | 0.090 | 0.206 | 0.661 |
| temperature_mean_origin | 38,416 | 19.165 | 6.845 | −1.313 | 28.878 |
| adults_origin | 38,416 | 23.126 | 83.544 | 0.062 | 891.457 |
| adults_s_origin | 38,416 | 11.515 | 43.661 | 0.005 | 545.809 |
| adults_u_origin | 38,416 | 11.611 | 44.958 | 0.013 | 417.970 |
| I_u_origin | 27,696 | 1619.118 | 2258.951 | 37.790 | 12,482.420 |
| I_s_origin | 27,696 | 14,930.430 | 18,537.480 | 427.439 | 115,032.800 |
| migration_stock | 50,338 | 6920.799 | 99,573.170 | 0.000 | 9,367,910.000 |
| distance | 55,138 | 5525.726 | 4379.361 | 1.000 | 19,147.900 |
| migration_flow | 24,144 | 4270.829 | 63,890.900 | 0.000 | 6,959,408.000 |
| migration_probability | 19,883 | 0.001 | 0.006 | 0.000 | 0.257 |
| migration_probability_u | 19,883 | 0.002 | 0.019 | 0.000 | 0.683 |
| migration_probability_s | 19,883 | 0.002 | 0.018 | 0.000 | 0.756 |
| temp_change_origin | 12,656 | 0.439 | 0.747 | −0.914 | 2.619 |

**Table A2.** Migration Gravity Model estimation (OLS).

| | Pooled | High-Skilled | Low-Skilled |
|---|---|---|---|
| $log(L_i)$ | −0.56 * | 0.17 | 0.47 |
| | (0.26) | (0.27) | (0.34) |
| $log(L_j)$ | −0.23 | −0.38 | −0.38 |
| | (0.26) | (0.28) | (0.34) |
| $log\left(\frac{GDPPC_j}{GDPPC_i}\right)$ | 0.52 *** | 0.17 | 0.41 * |
| | (0.14) | (0.26) | (0.16) |
| $log(d_{ij})$ | −0.08 *** | −0.06 *** | −0.06 *** |
| | (0.01) | (0.02) | (0.02) |
| $R^2$ | 0.11 | 0.13 | 0.13 |
| Adj. $R^2$ | 0.10 | 0.11 | 0.11 |
| Num. obs. | 25117 | 15629 | 15629 |

All estimations include origin and destination fixed effects. * $p$-value < 0.05, *** $p$-value < 0.001.

**Appendix C. OLG Model Description**

We developed a structural model of economic and demographic change based on an overlapping generation (OLG) framework [54,55]. The model includes two types of labor, a two-sector economy, and a large set of $I$ countries $j = 1, \ldots, I$. The simple nature of this model enables us to derive closed-form solutions that could be applied for the socioeconomic projections. One economic sector is agriculture (*a*), where only low-skilled labor (*u*) is employed [56,57]. The other sector is non-agriculture (*b*), and uses only high-skilled labor (*s*). We obtain the explicit form of the bilateral migration probabilities from the gravity model based on historical migration data. This model reflects the fact that migration decisions are based mainly on income considerations [58]. According to the results of our gravity model, the probability of an individual with skill level $k$ at time $t + 1$ to migrate from country $i$ to country $j$ is a function of the population of people with the same skill level $k$ in the sending and receiving and countries at time $t$ ($L_{i,t}^k$ and $L_{j,t}^k$, respectively), the

ratio of wages between the same type of labor in two countries at time $t$ ($\frac{w_{j,t}^k}{w_{i,t}^k}$), the distance

between the two countries ($d_{i,j}$), and the sending and receiving countries' fixed effects ($\eta_i$ and $\delta_j$):

$$\beta^k_{ij,t+1} = \theta^k_{ij} + \theta^k_1 log(L^k_{i,t}) + \theta^k_2 log(L^k_{j,t}) + \theta^k_3 log\left(\frac{w^k_{j,t}}{w^k_{i,t}}\right) + \theta^k_4 log(d_{ij}) + \eta_i + \delta_j, \qquad \text{(A1)}$$

where $\theta^k_{ij}$ is the migration residual between the two countries calculated from fitting the probability equation to the historical migration data from 1980 to 2000.

*Appendix C.1. Preferences*

An adult's utility comes from their level of consumption and from the future wages of their children:

$$v(c_t, n^s_t, n^u_t) = (1 - \gamma)ln(c_t) + \gamma\left[ln(n^s_t \mathbf{E}w^s_{t+1} + n^u_t \mathbf{E}w^u_{t+1})\right], \qquad \text{(A2)}$$

where $n^k_t$ is the number of children of skill level $k$, $c_t$ is consumption of a bundle of agricultural and non-agricultural goods, and $\mathbf{E}w^k_{t+1}$ represents the expected wages of children with skill level $k$ depending on where the children earn their income. The utility from the expected wages of children in country $i$ is therefore calculated as

$$ln(n^s_t \mathbf{E_s}w^s_{t+1} + n^u_t \mathbf{E_u}w^u_{t+1}) = ln(n^s_t \sum_{j=1}^{I} \beta^s_{ij,t+1} w^{j,s}_{t+1} + n^u_t \sum_{j=1}^{I} \beta^u_{ij,t+1} w^{j,u}_{t+1}), \qquad \text{(A3)}$$

For simplicity of these equations, we use country indices only when it is necessary to emphasise the difference between the countries. We normalize the price index of the consumption composite to one. Thus, the budget constraint corresponding to (A2) for every adult in each country is obtained by

$$c_t = \left(1 - \sum_{k=s,u} \tau^k_i n^k_t\right)w_t. \qquad \text{(A4)}$$

where $\tau^k_i$ is the fraction of time that a parent spends on raising a child of skill level $k$ in country $i$. We assume that raising high-skilled children is costlier than raising low-skilled children ($\tau^s > \tau^u$).

The maximization of (A2) subject to (A4) yields

$$c_t = (1 - \gamma)w_t \qquad \text{(A5)}$$
$$\sum_{k=s,u} \tau^k_i n^k_t = \gamma. \qquad \text{(A6)}$$

Equation (A6) encapsulates the quantity–quality trade-off. Because $\tau^s > \tau^u$ and the total time devoted to raising children is fixed (or increasing at a slower rate than population growth), individuals must decide between investing in a smaller number of children with higher skills and higher potential income or having a greater number of total children with lower skills and lower potential income. Therefore, in moving from an agricultural to a non-agricultural society each country is essentially substituting its cheap low-skilled labor with expensive high-skilled labor. This means that the trade-off between population growth and human capital development is embedded in the heart of the model, which can explain several of the inconsistencies between our results and the SSP projections.

In addition, for individuals in country $i$ to have both types of children, it must be the case that

$$\tau^r_i = \frac{\tau^s_i}{\tau^u_i} = \frac{\sum_{j=1}^{I} \beta^s_{ij,t} w^{j,s}_{t+1}}{\sum_{j=1}^{I} \beta^u_{ij,t} w^{j,u}_{t+1}}. \qquad \text{(A7)}$$

This equation shows that the ratio of the expected wages of children is equal to the ratio of child-rearing and migration costs.

*Appendix C.2. Consumption*

Total consumption by the labor of skill level $k$ is a constant elasticity of substitution (CES) function obtained by

$$c^k = \{\alpha(c_a^k)^{\frac{\epsilon-1}{\epsilon}} + (1-\alpha)(c_b^k)^{\frac{\epsilon-1}{\epsilon}}\}^{\frac{\epsilon}{\epsilon-1}}, \tag{A8}$$

where $\epsilon$ is the elasticity of substitution, $c_a$ is the consumption of agricultural goods, $c_b$ is the consumption of non-agricultural goods, and the time subscripts have been suppressed for convenience. As $\epsilon$ approaches zero, consumers receive less satisfaction from substituting non-agricultural goods for agricultural goods. At the limit, there is no substitution and the goods are consumed in fixed proportions. The consumer optimization problem conditioned on the budget constraint can be formulated using the Lagrangian multiplier $\lambda$:

$$Max\left\{c^k - \lambda\left(p_a c_a^k + p_b c_b^k - (1-\gamma)w^k\right)\right\}, \tag{A9}$$

where $p_b$ and $p_a$ are the prices of non-agricultural and agricultural goods, respectively. The solution to this optimization problem provides a relationship between these prices:

$$\frac{p_b}{p_a} = \left(\frac{1-\alpha}{\alpha}\right)\left(\frac{c_b^k}{c_a^k}\right)^{\frac{-1}{\epsilon}}, \tag{A10}$$

*Appendix C.3. Production*

We adopt a linear production function that captures the fact that agricultural production is relatively less skill-intensive [56,57]. In this respect, our model can therefore be seen as a simplified version of the sectoral migration model of [59]. Specifically,

$$Y_b = D_b(T)A_b L^s \tag{A11}$$
$$Y_a = D_a(T)A_a L^u, \tag{A12}$$

where $Y_{\varkappa}$, $\varkappa = a, b$ are outputs in sector $\varkappa$, and $L^k$, $k = s, u$ is total labor of skill level $k$. As the only endogenous factor in production is labor, within-country inequality is entirely driven by population composition, which is directly impacted by migration. The total factor of productivity (TFP) or technological change is defined as $A_{\varkappa}$, $\varkappa = a, b$ and $D_{\varkappa}(T)$ is the climate impact function for sector $\varkappa$ at temperature $T$. In order to analyze the effect of climate change in our model, we use RCP temperature projection data for each country (see the global projections in Figure A1) to calculate the sector-specific impact function:

$$D_{\varkappa}(T) = max\{g_{\varkappa,0} + g_{\varkappa,1}T + g_{k,2}T^2, D_{\varkappa}^{min}\}, \varkappa = a, b, \tag{A13}$$

where $g_{b,0} = 0.3$, $g_{b,1} = 0.08$ $g_{b,2} = -0.0023$, $g_{a,0} = -2.24$, $g_{a,1} = 0.308$, and $g_{a,2} = -0.0073$. The constant $D_{\varkappa}^{min}$ guarantees the minimum level of economic output at very high climate damages. For our analysis, we assume that $D_{\varkappa}^{min} = 10\%$. The damage function thus has the shape of a quadratic function with an optimal temperature between 17.4 (non-agricultural) and 21.1 (agriculture) degrees Celsius and with a maximum productivity loss of 90%. The shape of this function is depicted in Figure A1.

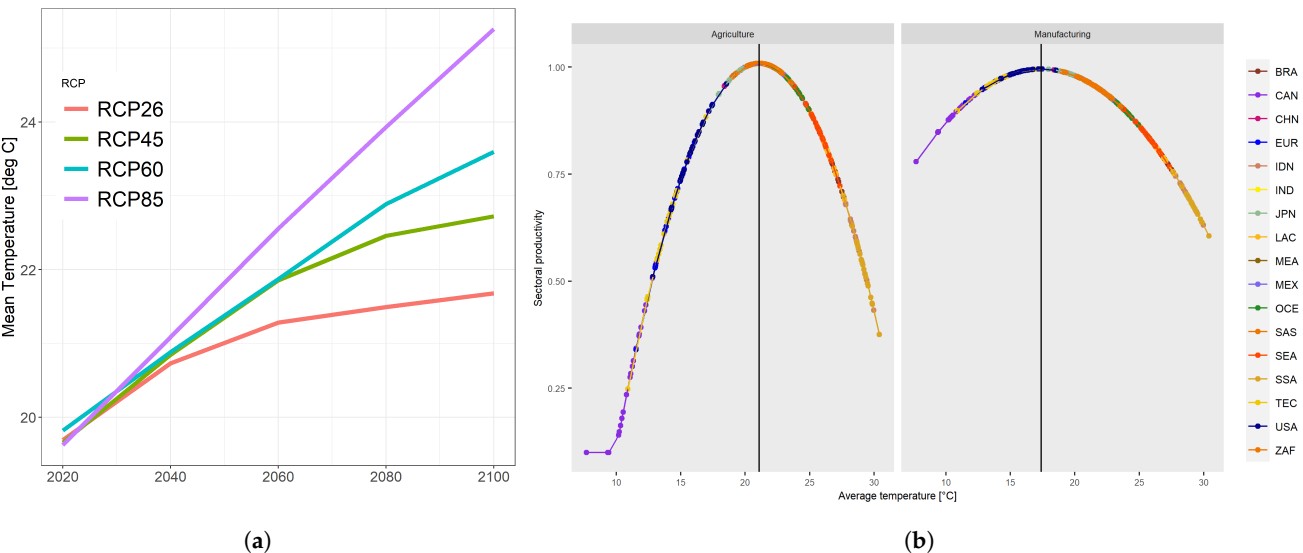

(**a**)                                        (**b**)

**Figure A1.** Global projections of temperature and climate change damage for agricultural and non-agricultural sectors (sectoral efficiency for different temperatures) based on calculations in [60]: (**a**) Global temperature projections; (**b**) Damage function.

Technological change evolves exogenously according to:

$$A_{\varkappa,t} = (1 + g_\varkappa)A_{\varkappa,t-1}, \ \varkappa = a, b. \tag{A14}$$

The total numbers of high-skilled and low-skilled workers are calculated by taking into account the possibility of labor movement to and from the country of interest. For example, the total number of laborers of skill level $k$ in a receiving country $j$ is:

$$L_{j,t+1}^k = \sum_{i=1}^{I} N_{i,t} \, n_{i,t}^k \, \beta_{ij,t+1}^k \tag{A15}$$

where $N_{i,t}$ is the adult population in country $i$ at time $t$. Wages can be calculated by taking the derivative of Equations (A11) and (A12):

$$
\begin{aligned}
w^s &= p_b \, D_b(T) \, A_b \\
w^u &= p_a \, D_a(T) \, A_a,
\end{aligned}
\tag{A16} \tag{A17}
$$

The consumption of good type $\varkappa$ in country $i$ by adults of each skill level is calculated using the following equations:

$$c_{i,\varkappa}^u = \frac{Y_{i,\varkappa}}{L_i^s \tau^r + L_i^u}, \qquad c_{i,\varkappa}^s = c_{i,\varkappa}^u \tau^r. \tag{A18}$$

*Appendix C.4. Inequality*

We can calculate the income inequality within each country by constructing a two-segment Lorenz curve based on empirical data. First, we obtain country quintiles ($\hat{q}_i^g$ for quintile $g$ in country $i$) from the SWIID database [61] and population $N_{k,i}$ and GDP data $Y_{k,i}$ from the World Development Indicators. Then, a five-segment Lorenz curve $LC$ based on the country quintiles is approximated by a two-segment curve where we compute the share of low-skilled labour $LC(L_i^u/N_i)$, resulting in the average wages $\hat{w}_i^u$ for low-skilled labor and $\hat{w}_i^s$ for high-skilled labor in country $i$ matching the GDP and the Lorenz curve at the point $(L_i^u/N_i)$. The inequality within a country can hence be expressed as the wage ratio of high-skilled to low-skilled labor ($\frac{w^s}{w^u}$). Inequality across countries is calculated for each skill level separately ($z^u$ and $z^s$ for low-skilled and high-skilled labor, respectively).

For example, for high-skilled labor it is the ratio of high-skilled wages in each country to the arithmetic average of high-skilled wages in all countries; therefore, for any country $i$ we have

$$z^u \quad = \quad \frac{w^u}{\left(\sum_{j=1}^{J} w_j^u\right)/J} \tag{A19}$$

$$\tag{A20}$$

$$z^s \quad = \quad \frac{w^s}{\left(\sum_{j=1}^{J} w_j^s\right)/J} \tag{A21}$$

*Appendix C.5. Equilibrium*

Combining Equations (A10), (A16), and (A17) with individual maximization and production yields the following equilibrium results for each country:

$$ln\left(\frac{L_{t+1}^s}{L_{t+1}^u}\right) = \epsilon\, ln\left(\frac{1-\alpha}{\alpha}\right) - \epsilon\, ln\left(\frac{w_{t+1}^s}{w_{t+1}^u}\right) - (1-\epsilon)\, ln\left(\frac{D_{b,t+1}(T)}{D_{a,t+1}(T)}\right) - (1-\epsilon)\, ln\left(\frac{A_{b,t+1}}{A_{a,t+1}}\right). \tag{A22}$$

Note that we have used the time index $t+1$ to highlight the fact that this equation is solved by parents (i.e., adults at time $t$) to calculate the number of their children (i.e., adults at time $t+1$). Therefore, the high-skilled and low-skilled number of children at time $t$ corresponds to $L_{t+1}^s$ and $L_{t+1}^u$ (the number of adults at time $t+1$). Solving Equations (A6) and (A22) together results in the optimal number and skill level of children. The only unknown in these equations is the ratio of future wages of the children $\frac{w_{t+1}^s}{w_{t+1}^u}$, which is in fact the expected wages of the children in the future and can be approximated by the current wage ratio of the parents at time $t$. Therefore, the equilibrium equation can be rewritten as

$$ln\left(\frac{n_t^s}{n_t^u}\right) = \epsilon\, ln\left(\frac{1-\alpha}{\alpha}\right) - \epsilon\, ln\left(\frac{w_t^s}{w_t^u}\right) - (1-\epsilon)\, ln\left(\frac{D_{b,t+1}(T)}{D_{a,t+1}(T)}\right) - (1-\epsilon)\, ln\left(\frac{A_{b,t+1}}{A_{a,t+1}}\right). \tag{A23}$$

Now, we can calculate the number of children of each skill level in each country for certain future climate and technological change trajectories. If an increase in temperature negatively affects agriculture more than manufacturing, then the ratio $ln\left(\frac{D_b(T)}{D_a(T)}\right)$ is an increasing function of temperature $T$. As the relative climate change impacts on agricultural productivity increase in developing countries, the price of agricultural goods increases [62,63]. In the absence of an explicit mechanism for trade between countries, the current model treats each country as a closed economy. Therefore, when substitution between goods is sufficiently low ($\epsilon < 1$) any increase in the relative price of agricultural output directly translates into an increase in the relative wages of low-skilled labor working in the agricultural sector. Without migration, this raises the relative return to work in agriculture, causing parents to have relatively more low-skilled children. However, when migration possibility is taken into account there is a parallel movement of human capital between countries from those with lower wages to those with higher wages. The interaction of these two movements within each country and between countries defines the optimal level of population at end of each period.

*Appendix C.6. Calibration*

The model was calibrated with SSP projection data on the ratio of high-skilled to low-skilled labor in the 21st century. We chose the parameters to match demographic projections for each country. Historical data on migration were used to calibrate the migration probabilities.

We use several parameter values presented in [60] for other equations in the OLG model. We take $\epsilon = 0.75$ and $\alpha = 0.55/2$. We normalize the total time spent on raising children to $\gamma_0 = 45\%$ of total adult time in the year 1980 for all countries. We assume that

goods are complementary, and therefore the values of the elasticity of substitution remain below one.

We calibrated the model to find the ratio of productivities in the beginning and end years, 2000 and 2040, as well as $\tau^s$, $\tau^u$, $g_b$, and $g_a$. To do this, for each country we used historical population data in the year 2000 (i.e., population growth rate between 1980 and 2000, $r_{2000}$ along with the skill ratio in 2000, $h_{2000}$) and the projected skill ratio in 2040 ($h_{2040}$) from the Wittgenstein Centre [64]. Figure A2 shows the underlying assumptions in the SSP scenarios with regard to adult population, skill ratio, GDP per capita, and Gini index throughout the century. In this paper, we only used the country-level projections of skill ratio for calibrating our model.

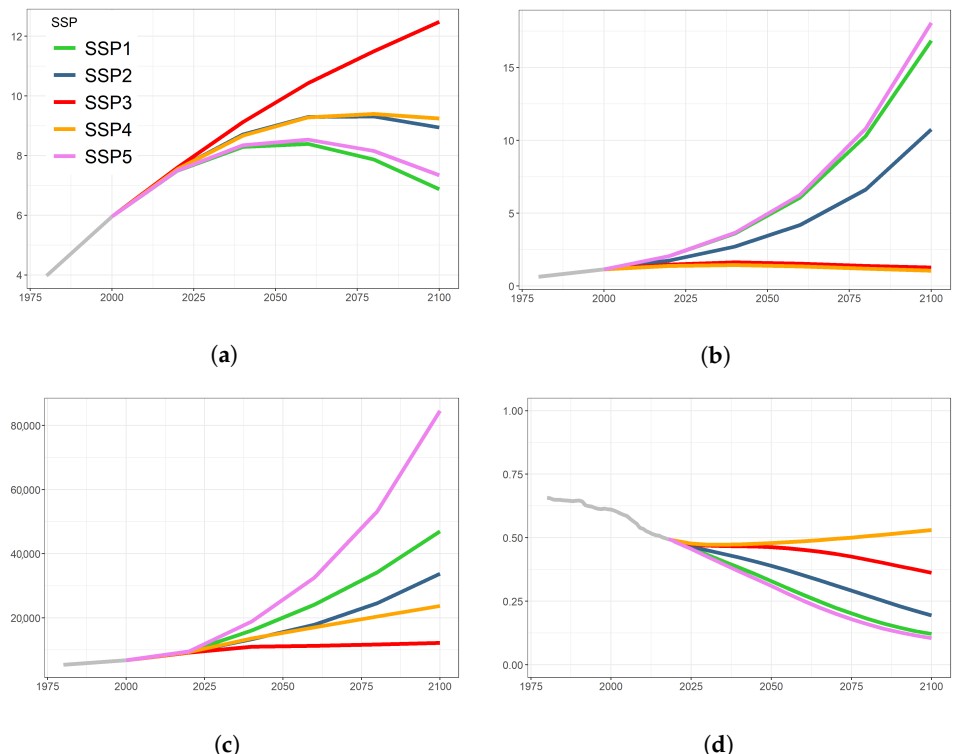

**Figure A2.** Global socioeconomic indicators under different SSP scenarios based on data from [37]: (**a**) Global adult population; (**b**) High-skilled to Low-skilled ratio; (**c**) GDP per capita; (**d**) Gini index.

Even though we did not directly use SSP migration projections in our model, they were already taken into account in the calculation of the future skill ratios. To reduce the impact of this bias in the calibration of our model, we devised two remedies. First, instead of following the skill ratio trajectory in all intermediate points, we only used a single projection point for our calibration. Second, instead of using the end-of-the-century point (year 2100) we used a middle-of-the-century point (year 2040) as our projection point. In this way, we allow the model to be less constrained by SSP migration-induced biases in the second half of the century. Although a perfect remedy would be to correct the projected skill ratio for the migration numbers in each SSP, the available reported SSP migration projections are total numbers without skill differentiation [65], and therefore the full neutralization of migration data would require additional assumptions about the skill ratio of migration flows.

We treat anyone with a high school education and higher as high-skilled labor. Moreover, we computed initial wages by constructing a Gini coefficient for each country and, based on high- and low-skilled labor population shares, we computed the relative wages to match the regional empirical Gini coefficient in 2000 (based on the World Development Indicators). To construct the empirical Gini coefficient, we used the skill ratio of each country in year 2000 ($h_{2100}$) to find the wages of high-income and low-income sections of the

society. For example, if the skill ratio in 2000 is $h_{2000} = \frac{2}{3}$, we assume this will be translated to $\psi_0^u = \frac{1}{1+h_{2000}} = 60\%$ share of the low-income population and $\psi_0^s = \frac{h_{2000}}{1+h_{2000}} = 40\%$ share of the high-income population. The average wages of each group were calculated using quintile data from the World Bank. The respective wages of low-skilled and high-skilled labor are calculated from the equations below:

$$w_0^u = \frac{\sum_{i=1}^{5 \times \psi_0^u} q_i \times Y_{2000}}{N_{2000}}, \tag{A24}$$

$$w_0^s = \frac{\sum_{i=5 \times \psi_0^u+1}^{5} q_i \times Y_{2000}}{N_{2000}}. \tag{A25}$$

where $q_i$ is the income quintile $i$, $Y_{2000}$ is the GDP, and $N_{2000}$ is the adult population in year 2000. For each country, we assume that in the year 2000, Equation (A7) holds in its simplified form $\frac{\tau^s}{\tau^u} = \frac{w^s}{w^u} = w_{r0}$ where $w_{r0}$ is the initial wage ratio of high-skilled to low-skilled labor. We solve the following two equations to obtain the time cost of raising children, $(\tau^u)$ and $(\tau^s)$:

$$\tau^u = \gamma \frac{1 + h_{2000}}{(1 + r_{2000})(1 + h_{2000} \times w_{r0})}, \tag{A26}$$

$$\tau^s = \tau^u \times w_{r0}. \tag{A27}$$

where $r_{2000}$ is the adult population growth rate between the years 1980 and 2000. Next, we use Equation (A23) to solve for the ratio of the initial and final technology levels, $A_{r,2000} = \frac{A_{b,2000}}{A_{a,2000}}$ and $A_{r,2040} = \frac{A_{b,2040}}{A_{a,2040}}$, assuming that the ratio of climate impacts is fixed at its year 2000 level $(D_{r0} = \frac{D_{b,2000}}{D_{a,2000}})$ and the wage ratios stay at $w_{r0}$ level for each country. The skill ratio in 2040 is taken from each SSP projection for each country.

$$ln(A_{r,2000}) = \frac{\epsilon}{1-\epsilon} ln(\frac{1-\alpha}{\alpha}) - \frac{\epsilon}{1-\epsilon} ln(w_{r0}) - ln(D_{r0}) - (1-\epsilon) ln(h_{2000}) \tag{A28}$$

$$ln(A_{r,2040}) = \frac{\epsilon}{1-\epsilon} ln(\frac{1-\alpha}{\alpha}) - \frac{\epsilon}{1-\epsilon} ln(w_{r0}) - ln(D_{r0}) - (1-\epsilon) ln(h_{2040}). \tag{A29}$$

Finally, we find the technology growth rates. By assumption, the growth rate of $\frac{A_b}{A_a}$ is constant:

$$\frac{A_{b,2040}}{A_{a,2040}} = (1 + g_r)^{\frac{(2040-2000)}{20}} \frac{A_{b,2000}}{A_{a,2000}}, \tag{A30}$$

where $g_r$ is the growth rate of the technology ratio. It is the only unknown variable in this equation, and satisfies

$$1 + g_r = \frac{1 + g_b}{1 + g_a}, \tag{A31}$$

where $g_b$ is the growth rate of $A_b$ and $g_a$ is the growth rate of $A_a$. Noting that large developed countries which have nearly all of their production in manufacturing grow at a much higher rate, we set $g_b = 0.1$ per year for developed countries and $g_b = 0.001$ per year for developing countries. Now, $g_a$ can be extracted from Equation (A31).

*Appendix C.7. Solution Algorithm*

We can solve the model using a series of dynamic equations in order. First, given the temperature projections of each RCP scenario we can calculate the damages from Equation (A13). Next, we calculate the exogenous component of technology using Equation (A14). All of the economic decisions are captured by Equation (A23), which can now be solved for the ratio of high-skilled to low-skilled individuals in every period. We can then solve for the level of the population such that Equation (A6) holds (we assume

an increase of 5% in child-rearing costs for both types of children globally in each period based on the evidence suggesting an increase in parenting time in selected countries [66]).

The OLG model solution algorithm can be summarized as follows:

1.  Calibrate the exogenous growth rate of productivity for each country based on the year 2000 population and the year 2040 projected skill ratio (Equations (A26) to (A31))
2.  Run the model through time from 2000 to 2100 with 20-year intervals

    (a)  Calculate the number of children and their education levels for all countries. For each country, solve the parents' utility optimization problem given the current wage ratios of high-skilled to low-skilled labor and the projections of technology growth and climate impacts in different sectors (Equations (A6) and (A23))

    (b)  Calculate the bilateral migration probabilities for every pair of countries using their current population, wages, and distance according to Equation (A1).

    (c)  Redistribute the next generation's population calculated in step (a) according to the migration probabilities obtained in step (b).

    (d)  Update the next generation's population and skill ratio and calculate the economic output and wages for the next time step as follows:

$$ln(\frac{w_{t+1}^s}{w_{t+1}^u}) = ln(\frac{1-\alpha}{\alpha}) - \frac{1}{\epsilon} ln(\frac{L_{t+1}^s}{L_{t+1}^u}) - \frac{1-\epsilon}{\epsilon} ln(\frac{D_{b,t+1}(T)}{D_{a,t+1}(T)}) - \frac{1-\epsilon}{\epsilon} ln(\frac{A_{b,t+1}}{A_{a,t+1}}). \tag{A32}$$

These wages will be used to calculate the probability of migration in the next period as well as the utility optimization that the parents in next generation use to find the optimal skill ratio of their children (Equation (A23)). Therefore, the impacts of migration on local wages can influence the wage inequality and migration patterns in subsequent periods.

## Appendix D. Alternative Setups

In this section, we report the results of several modifications to the model and its assumptions, along with robustness checks and sensitivity analyses.

### Appendix D.1. PPML Migration Probability Estimation

The PPML model replicates the probability estimation based on [9,29].

$$\frac{M_{ij,t+1}^k}{L_{i,t+1}} = exp\left[\theta_1^k log(L_{i,t}^k) + \theta_2^k log(L_{j,t}^k) + \theta_3^k log\left(\frac{GDPPC_{j,t}^k}{GDPPC_{i,t}^k}\right) + \theta_4^k log(d_{ij}) + \eta_i + \delta_j\right]\varepsilon_{ij} \tag{A33}$$

The results of this alternative setup are presented in Table A3. Although this model is better at dealing with historical migration panel data that contain large amount of zero values and provides a better estimate of historical migration trends, its application in the OLG model for estimating future migration probabilities is very limited. The main reason for this is the exponential function on the right hand-side, which inflates any changes in the scale of the variables and radically increases the probability on the left hand-side.

For example, when running the OLG model under RCP2.6 and SSP1 (very mild inequality and climate conditions), we observe that the exponential form of the probability function has a large impact on the migration probabilities of smaller countries in later years during the simulation. We took high-skilled (*h*) migration from Colombia to Spain and to the USA as an example and investigating how these migration probabilities evolve over time; the results of our analysis are shown in Table A4. As is shown, the high-skilled migration probability to Spain increases drastically from about 0.2% in the 2000–2020 period to nearly 34.7% in the last period. Meanwhile, the high-skilled migration probability from Colombia to the USA starts at around 1.1% and reaches about 78.4%. As a result of just these two migration probabilities, the total migration probability exceeds one and makes the model's calculation of population and migration flows infeasible. Therefore, we ruled out the PPML model as an appropriate estimation model for our migration projection, and

instead used the OLS model with logarithmic terms, described in the manuscript as the main estimation model. The results of the OLS calculation for the same example are shown in Table A5.

**Table A3.** Migration Gravity Model estimation (PPML).

|  | **Pooled** | **High-Skilled** | **Low-Skilled** |
|---|---|---|---|
| $log(L_i)$ | −0.23 | −0.20 | 0.80 ** |
|  | (0.17) | (0.18) | (0.25) |
| $log(L_j)$ | −1.51 *** | −1.01 *** | −0.09 |
|  | (0.22) | (0.23) | (0.18) |
| $log\left(\frac{GDPPC_j}{GDPPC_i}\right)$ | 0.75 *** | 0.63 ** | 0.34 *** |
|  | (0.10) | (0.22) | (0.08) |
| $log(d_{ij})$ | −0.24 *** | −0.24 *** | −0.24 *** |
|  | (0.01) | (0.01) | (0.01) |
| Num. obs. | 25,117 | 15,629 | 15,629 |

All estimations include origin and destination fixed effects. ** $p$-value < 0.01, *** $p$-value < 0.001.

**Table A4.** Probability of high-skilled (*h*) migration from Colombia to Spain (in black) and to the USA (in blue) with PPML estimation.

| Period | Destination | Constants | $\theta_1^h L_i^h$ | $\theta_2^h L_j^h$ | $\theta_3^h \frac{w_j^h}{w_i^h}$ | Sum | Probability |
|---|---|---|---|---|---|---|---|
| 2000–2020 | Spain | 20.215 | −3.279 | −17.009 | 0.759 | 0.686 | 0.198% |
|  | USA | 24.078 | −3.279 | −19.259 | 0.889 | 2.429 | 1.135% |
| 2020–2040 | Spain | 20.215 | −3.387 | −17.208 | 1.971 | 1.591 | 0.491% |
|  | USA | 24.078 | −3.387 | −19.491 | 2.057 | 3.257 | 2.597% |
| 2040–2060 | Spain | 20.215 | −3.397 | −17.125 | 3.163 | 2.856 | 1.739% |
|  | USA | 24.078 | −3.397 | −19.663 | 3.236 | 4.254 | 7.039% |
| 2060–2080 | Spain | 20.215 | −3.342 | −16.917 | 4.353 | 4.309 | 7.437% |
|  | USA | 24.078 | −3.342 | −19.761 | 4.422 | 5.397 | 22.074% |
| 2080–2100 | Spain | 20.215 | −3.213 | −16.699 | 5.545 | 5.848 | 34.654% |
|  | USA | 24.078 | −3.213 | −19.814 | 5.614 | 6.665 | 78.446% |

**Table A5.** Probability of high-skilled (*h*) migration from Colombia to Spain (in black) and to the USA (in blue) with OLS estimation.

| Period | Destination | Constants | $\theta_1^h L_i^h$ | $\theta_2^h L_j^h$ | $\theta_3^h \frac{w_j^h}{w_i^h}$ | Sum | Probability |
|---|---|---|---|---|---|---|---|
| 2000–2020 | Spain | 8.547 | 2.819 | −6.438 | 0.204 | 5.132 | 0.513% |
|  | USA | 18.287 | 2.819 | −7.289 | 0.239 | 14.056 | 1.406% |
| 2020–2040 | Spain | 8.547 | 2.908 | −6.519 | 0.529 | 5.465 | 0.547% |
|  | USA | 18.287 | 2.908 | −7.390 | 0.553 | 14.358 | 1.436% |
| 2040–2060 | Spain | 8.547 | 2.910 | −6.500 | 0.849 | 5.806 | 0.581% |
|  | USA | 18.287 | 2.910 | −7.470 | 0.869 | 14.596 | 1.460% |
| 2060–2080 | Spain | 8.547 | 2.864 | −6.440 | 1.169 | 6.140 | 0.614% |
|  | USA | 18.287 | 2.864 | −7.515 | 1.187 | 14.823 | 1.482% |
| 2080–2100 | Spain | 8.547 | 2.786 | −6.358 | 1.489 | 6.464 | 0.646% |
|  | USA | 18.287 | 2.786 | −7.532 | 1.507 | 15.048 | 1.505% |

*Appendix D.2. Migration Policy*

SSP scenarios provide different narratives about international migration [67] that cannot be reflected in our OLG model without introducing additional assumptions. For example, high inequality assumptions in the SSP3 and SSP4 scenarios translate into high

bilateral migration flows following Equation (A1). However, these SSP scenarios have strong assumptions about regional rivalries and stricter migration policies which cannot be captured by this model. In order to adjust the scale of international migration flows and make them consistent with SSP narratives, one can introduce an ad hoc policy parameter which applies universally to all bilateral migration probabilities. Thus, we can modify the migration probability equation to account for underlying migration policy differences between SSP scenarios:

$$\beta^k_{ij,t+1} = \left[ \theta^k_{ij} + \theta^k_1 log(L^k_{i,t}) + \theta^k_2 log(L^k_{j,t}) + \theta^k_3 log\left(\frac{w^k_{j,t}}{w^k_{i,t}}\right) + \theta^k_4 log(d_{ij}) + \eta_i + \delta_j \right] \times (1 - \zeta^{ssp}_{ij}), \qquad \text{(A34)}$$

where parameter $0 \leq \zeta^{ssp}_{ij} \leq 1$ is the relative cost of migration of labor of skill level $k$ from country $i$ to $j$ reflecting the restrictiveness of policies that govern the migration of labor from country $i$ to $j$ under a given *ssp* scenario. When this parameter is zero, we assume no restriction in labor mobility, while when the value is equal to 1 it means absolute closure of borders. Within the SSP framework and for all pairs of countries, we assume $\zeta^{ssp}_{ij} = 0.5$ for SSP1, SSP2, and SSP4 where international migration is *medium*, $\zeta^{ssp}_{ij} = 0.1$ for SSP5 where international migration is *high*, and $\zeta^{ssp}_{ij} = 0.9$ for SSP3 where international migration is *low*. Figure A3 shows the impact of the introduction of this parameter in the model on total migration flows across SSP scenarios.

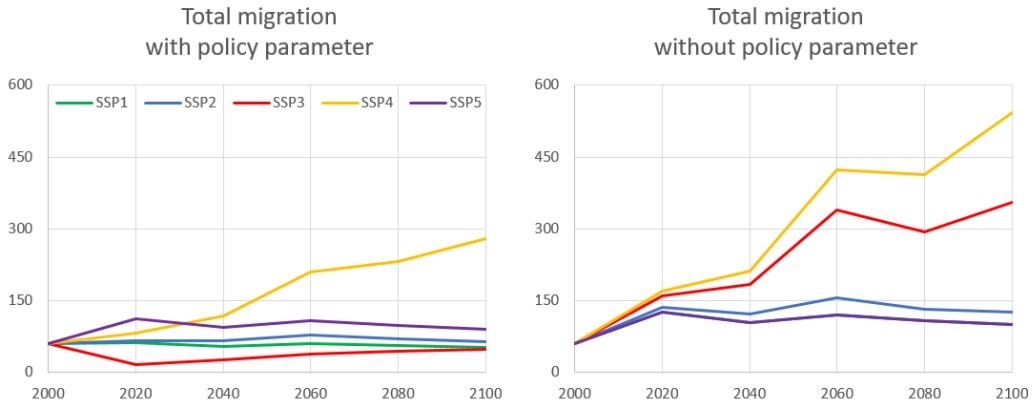

**Figure A3.** The impact of the policy parameter on total migration flows across SSP scenarios.

As expected, the introduction of $\zeta^{ssp}_{ij} = 0.5$ reduces migration flows by about 50% under the SSP1, SSP2, and SSP4 scenarios, while migration flows under SSP5 remain almost unchanged. On the other hand, the introduction of $\zeta^{ssp}_{ij} = 0.9$ for SSP3 reduces migration flows drastically and results in an early collapse of migration flows in 2020.

*Appendix D.3. RCP2.6 Scenario*

In order to distinguish the impact of climate change on international migration under each SSP, we considered two RCP cases with different temperature trajectories (see panel (a) in Figure A1). For the *baseline case*, we allowed for temperatures to change according to the RCP6.0 projections; therefore, migration is affected indirectly by climate change through its impact on wages. The second case is based on a low-risk climate change scenario, RCP2.6, where we assumed that climate change is limited and migration is marginally impacted. The results of the *baseline case* are presented in the main manuscript. Here we provide additional results mainly based on the RCP2.6 case.

First, we consider the total migration flows over time presented in panel (a) in Figure A4. The patterns are similar to the *baseline case* in the main manuscript. In both cases, total migration in the SSP1, 2, and 5 scenarios demonstrates an initial increase through the middle of the century and a downward trend afterwards. In the SSP3 and SSP4 scenarios,

however, widespread inequality drives international migration flows to their peak by the end of the century. The global flow of high-skilled migrants follows similar pattern for all the SSP scenarios, as shown in panel (b) in Figure A4. However, the global flow of low-skilled migrants decreases under SSP1, 2, and 5 scenarios while increasing under the SSP3 and SSP4 scenarios, as shown in panel (c) in Figure A4.

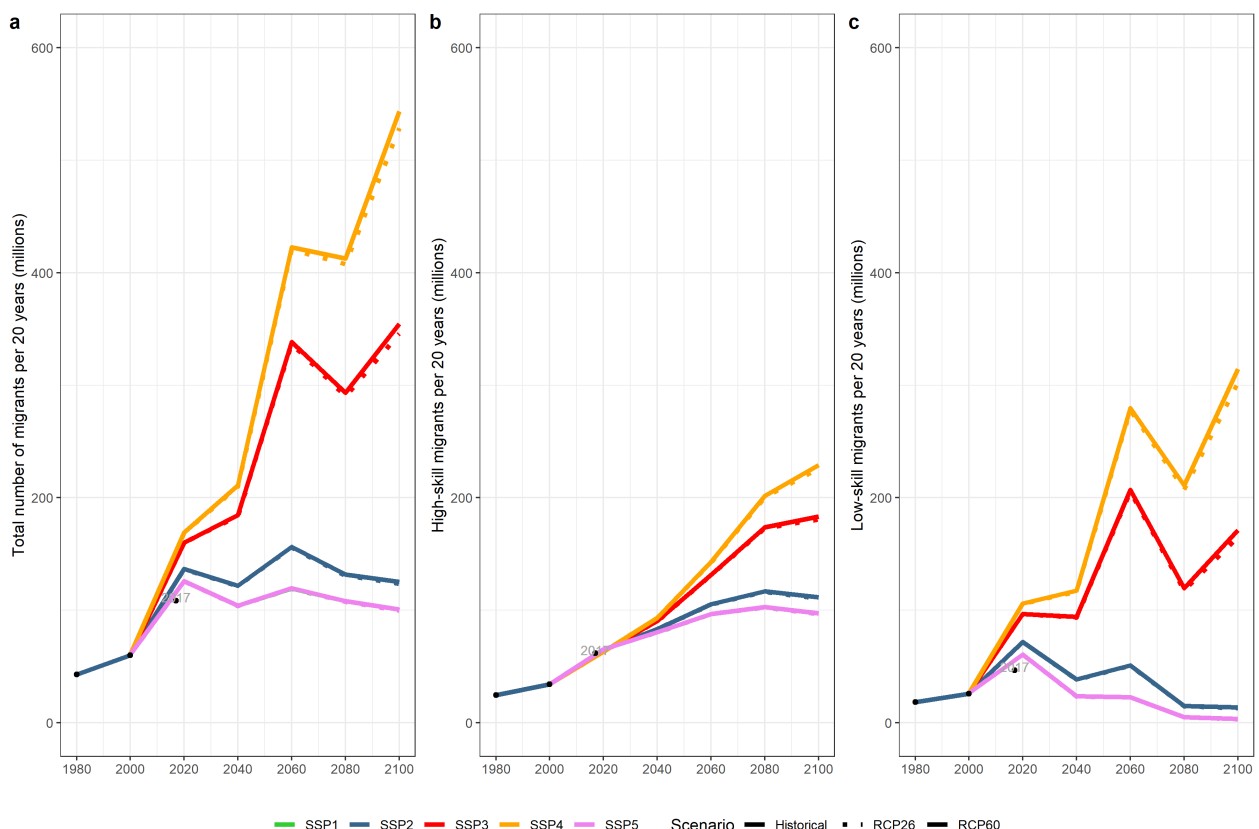

**Figure A4.** Historical and projected twenty-year global migration flows for RCP2.6 and RCP6.0 cases: (**a**) Total global migration; (**b**) High-skilled global migration; (**c**) Low-skilled global migration.

In terms of regional hot spots, India emerges as a major sending country while Europe keeps its position as an attractive destination and the role of the US as a major receiving country declines. Figure A5 shows the breakdown of migration flows between inside regions and across SSPs for the RCP2.6 case.

Figures A6 and A7 show the breakdown of total migration flows in 2100 by region and across SSP scenarios for the RCP26 case and *baseline case*, respectively.

Panel (a) in Figure A8 shows the change in human capital accumulation across the SSPs for the RCP2.6 case. In SSP1, SSP2, and SSP5, the population of high-skilled labor compared to low-skilled labor increases dramatically towards the end of the century due to a global education push. In contrast, the ratio of high-skilled to low-skilled labor does not increase as quickly in SSP3 and SSP4. Panel (b) in Figure A8 shows the percentage change in the high-skilled, low-skilled, and total adult population in the *baseline case* compared to the RCP2.6 case. In all scenarios, low-skilled labor shows a fast increasing trend while high-skilled labor increases at a much slower pace.

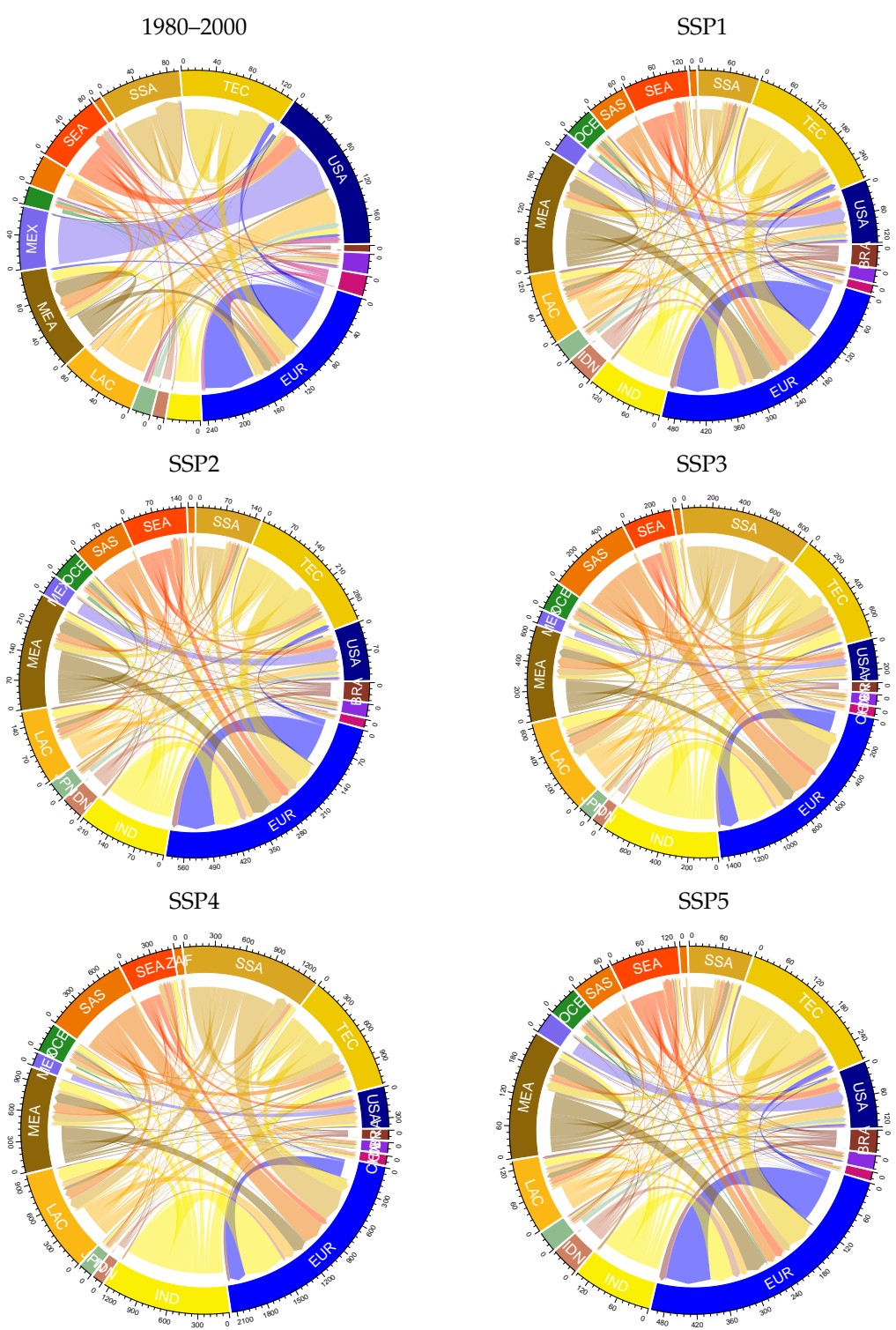

**Figure A5.** Regional bilateral migration flows between 2080 and 2100, annually in 100 thousands under RCP2.6.

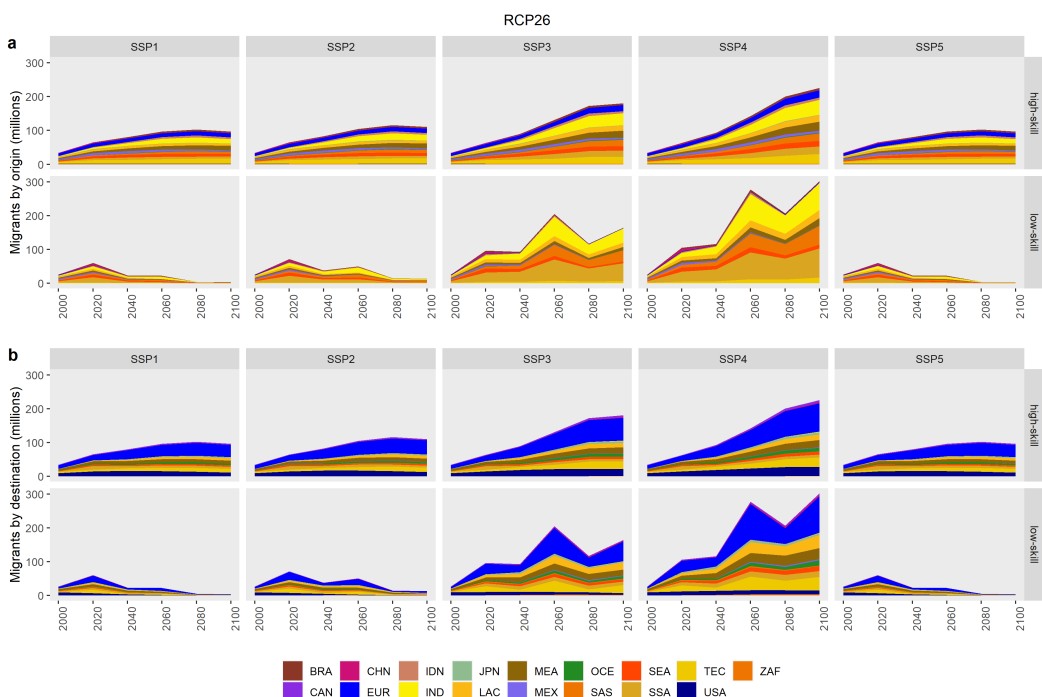

**Figure A6.** Migration flows of high-skilled and low-skilled labor under RCP2.6 case: distribution of migrants by (**a**) origin and (**b**) destination.

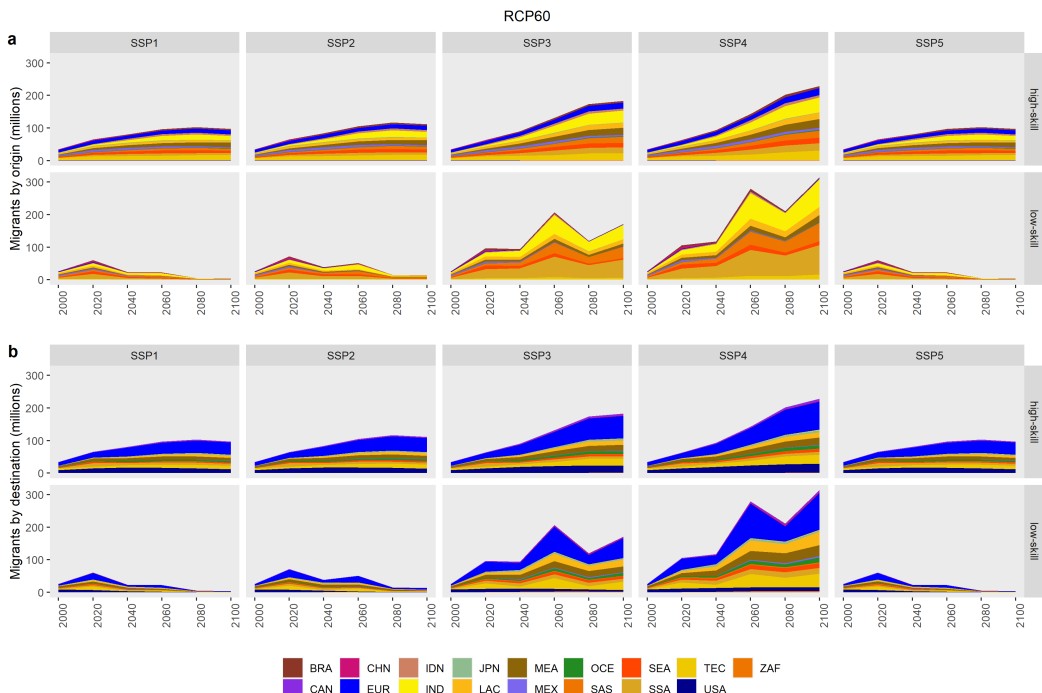

**Figure A7.** Migration flows of high-skilled and low-skilled labor under RCP6.0 case: distribution of migrants by (**a**) origin and (**b**) destination.

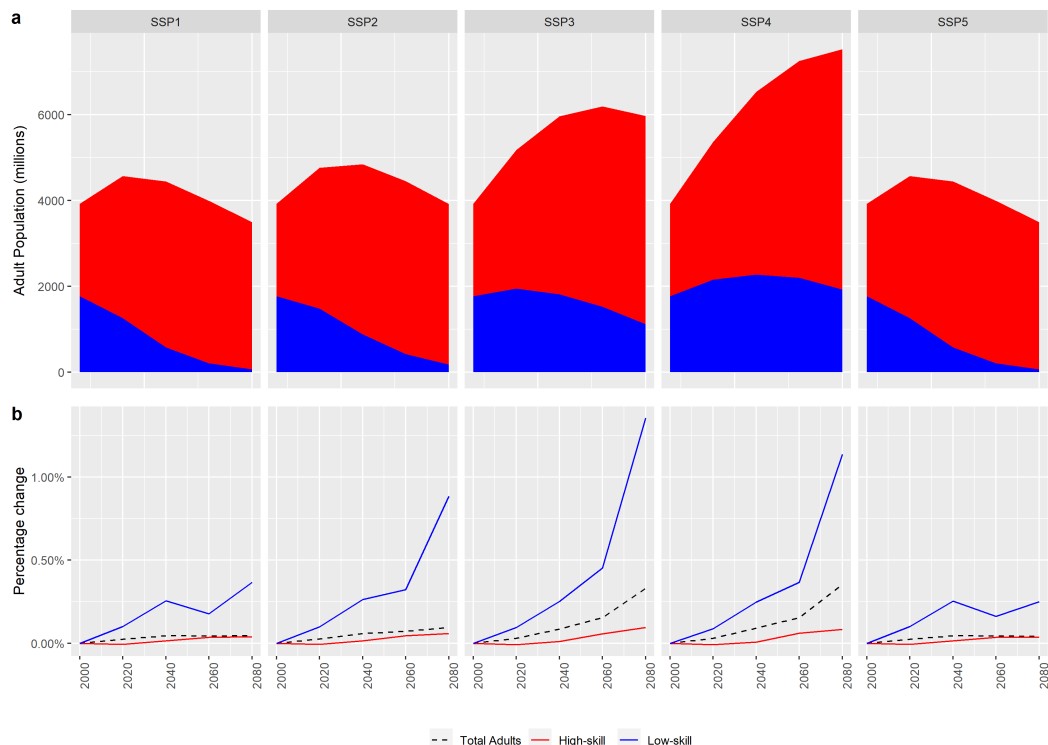

**Figure A8.** Human capital accumulation across the SSPs: (**a**) High-skilled and low-skilled adult population under RCP2.6 case; (**b**) Percentage change in high-skilled (red colour), low-skilled (blue colour), and total adult population (dotted line) under RCP6.0 case compared to RCP2.6 case.

*Appendix D.4. Climate Change Foresight*

Climate change foresight is one of the key determinants of fertility and child education decisions, as shown by the relative productivity ratio ($\frac{D_{b,t+1}(T)}{D_{a,t+1}(T)}$) in Equation (A22). Although the underlying assumption here is that the parents can foresee upcoming climate changes and internalize such signals in their fertility decisions, it might be hard to justify the reality of such an assumption in the real world. In order to check the robustness of our results against this assumption, we modified the model to use the parents' realization of the productivity ratio due to climate damages at the current time ($\frac{D_{b,t}(T)}{D_{a,t}(T)}$) instead of the projected ratio in the next period. We ran the model under RCP2.6 and RCP6.0 climate change scenarios. The results for the total population, global skill ratio, and global migration flows are reported in three panels of Table A6 as percentage changes compared to the original case with perfect foresight. In the case without climate change foresight, parents in developing countries with larger agricultural sectors underestimate the future warming, and therefore have less incentive to have a higher number of low-skilled children compared to the original case with perfect climate change foresight. Therefore, the overall result is a lower population, lower migration, and higher skill ratio globally in most years for most SSPs. The difference between these two cases in terms of total global population and migration flows are less than 1% in all SSP-RCP scenarios throughout the century. Although the differences in skill ratios are higher, they are under 2% in all SSP-RCP scenarios. Comparing the differences between the two RCP scenarios (RCP2.6 with lower and RCP6.0 with higher climate change) reveals that the difference between the case with perfect climate foresight and the case without climate change foresight is larger under RCP6.0, as climate change impacts are larger in this scenario.

**Table A6.** Percentage change in total (adult) population, skill ratio, and total migration under RCP2.6 (in black) and RCP6.0 (in blue) scenarios in the case without climate change foresight compared to the case with climate change foresight.

| Percentage Change in Total Population | | SSP1 | SSP2 | SSP3 | SSP4 | SSP5 |
|---|---|---|---|---|---|---|
| 2000 | RCP2.6 | 0.00% | 0.00% | 0.00% | 0.00% | 0.00% |
|  | RCP6.0 | 0.00% | 0.00% | 0.00% | 0.00% | 0.00% |
| 2020 | RCP2.6 | −0.21% | −0.22% | −0.23% | −0.28% | −0.21% |
|  | RCP6.0 | −0.24% | −0.25% | −0.26% | −0.31% | −0.24% |
| 2040 | RCP2.6 | −0.15% | −0.15% | −0.12% | −0.12% | −0.15% |
|  | RCP6.0 | −0.20% | −0.21% | −0.20% | −0.22% | −0.20% |
| 2060 | RCP2.6 | −0.17% | −0.16% | −0.11% | −0.11% | −0.17% |
|  | RCP6.0 | −0.19% | −0.21% | −0.23% | −0.23% | −0.19% |
| 2080 | RCP2.6 | −0.17% | −0.16% | −0.06% | −0.07% | −0.17% |
|  | RCP6.0 | −0.19% | −0.22% | −0.33% | −0.35% | −0.19% |
| 2100 | RCP2.6 | −0.17% | −0.16% | −0.01% | 0.02% | −0.17% |
|  | RCP6.0 | −0.20% | −0.26% | −0.47% | −0.56% | −0.20% |
| **Percentage Change in Skill Ratio** | | **SSP1** | **SSP2** | **SSP3** | **SSP4** | **SSP5** |
| 2000 | RCP2.6 | 0.00% | 0.00% | 0.00% | 0.00% | 0.00% |
|  | RCP6.0 | 0.00% | 0.00% | 0.00% | 0.00% | 0.00% |
| 2020 | RCP2.6 | 0.84% | 0.78% | 0.66% | 0.78% | 0.84% |
|  | RCP6.0 | 0.95% | 0.89% | 0.76% | 0.87% | 0.95% |
| 2040 | RCP2.6 | −0.67% | −0.59% | −0.50% | −0.62% | −0.67% |
|  | RCP6.0 | −0.40% | −0.34% | −0.28% | −0.38% | −0.41% |
| 2060 | RCP2.6 | −0.19% | −0.20% | −0.18% | −0.19% | −0.19% |
|  | RCP6.0 | −0.13% | −0.01% | 0.15% | 0.06% | −0.15% |
| 2080 | RCP2.6 | 0.04% | −0.28% | −0.54% | −0.41% | 0.07% |
|  | RCP6.0 | 0.24% | 0.44% | 0.67% | 0.61% | 0.18% |
| 2100 | RCP2.6 | −0.11% | −0.60% | −0.98% | −0.82% | 0.04% |
|  | RCP6.0 | 1.37% | 1.34% | 1.19% | 1.13% | 1.15% |
| **Percentage Change in Total Migration** | | **SSP1** | **SSP2** | **SSP3** | **SSP4** | **SSP5** |
| 2000 | RCP2.6 | 0.00% | 0.00% | 0.00% | 0.00% | 0.00% |
|  | RCP6.0 | 0.00% | 0.00% | 0.00% | 0.00% | 0.00% |
| 2020 | RCP2.6 | −0.08% | −0.07% | −0.06% | −0.10% | −0.08% |
|  | RCP6.0 | −0.12% | −0.12% | −0.10% | −0.14% | −0.12% |
| 2040 | RCP2.6 | 0.03% | 0.05% | 0.09% | 0.11% | 0.03% |
|  | RCP6.0 | −0.05% | −0.06% | −0.08% | −0.07% | −0.05% |
| 2060 | RCP2.6 | −0.15% | −0.24% | −0.46% | −0.51% | −0.14% |
|  | RCP6.0 | −0.24% | −0.39% | −0.71% | −0.75% | −0.23% |
| 2080 | RCP2.6 | −0.02% | −0.01% | 0.03% | 0.01% | −0.01% |
|  | RCP6.0 | −0.11% | −0.19% | −0.49% | −0.61% | −0.11% |
| 2100 | RCP2.6 | −0.03% | −0.03% | −0.08% | −0.05% | −0.03% |
|  | RCP6.0 | −0.19% | −0.36% | −1.03% | −1.18% | −0.17% |

## Appendix E. Comparison with SSP Migration

Table A7 shows the comparison between the results of our model and the underlying migration assumptions in the SSPs for selected countries [65] for the period 2060–2080. The numbers reflect the migration stress defined as the net number of migrants in the

20-year period as a percentage of the total adult population in 2080. The OLG results are generally higher than the SSP assumptions; however, the migration signs are mostly consistent between our model and the SSP underlying assumptions.

**Table A7.** Comparison of the SSP Migration assumptions from [65] (in black) with the results of the OLG model (in blue) for selected countries. The values show the 20-year net migration stress (percentage of net migration in total adult population) for the period 2060–2080.

| Country | | SSP1 | SSP2 | SSP3 | SSP4 | SSP5 |
|---|---|---|---|---|---|---|
| **Brazil** | **SSP** | −0.33 | 0.34 | −0.15 | −0.25 | −0.66 |
| | **OLG** | −7.94 | −7.59 | −6.91 | −7.41 | −7.94 |
| **China** | **SSP** | −0.14 | −0.15 | −0.07 | −0.11 | −0.29 |
| | **OLG** | 0.07 | 0.12 | 0.52 | 0.57 | 0.07 |
| **Egypt** | **SSP** | −0.60 | −0.60 | −0.25 | −0.46 | −1.20 |
| | **OLG** | −9.16 | −8.65 | −7.32 | −7.52 | −9.16 |
| **Germany** | **SSP** | 3.36 | 3.75 | 2.53 | 4.26 | 4.60 |
| | **OLG** | 7.25 | 8.01 | 12.20 | 15.30 | 7.21 |
| **India** | **SSP** | −0.30 | −0.30 | −0.13 | −0.24 | −0.59 |
| | **OLG** | −5.42 | −5.61 | −6.75 | −7.63 | −5.42 |
| **Indonesia** | **SSP** | −0.70 | −0.69 | −0.29 | −0.52 | −1.39 |
| | **OLG** | −7.78 | −7.38 | −6.38 | −6.52 | −7.78 |
| **Mexico** | **SSP** | −2.31 | −2.32 | −0.96 | −1.80 | −4.60 |
| | **OLG** | −9.41 | −8.92 | −5.32 | −4.90 | −9.42 |
| **Nigeria** | **SSP** | −0.45 | −0.44 | −0.18 | −0.40 | −0.90 |
| | **OLG** | −6.09 | −5.64 | −5.58 | −5.91 | −6.09 |
| **Pakistan** | **SSP** | −1.79 | −1.80 | −0.75 | −1.69 | −3.58 |
| | **OLG** | −6.95 | −7.40 | −10.31 | −11.69 | −6.95 |
| **Korea** | **SSP** | 0.31 | 0.35 | 0.24 | 0.40 | 0.43 |
| | **OLG** | −0.50 | 0.07 | 3.16 | 4.77 | −0.51 |
| **Russia** | **SSP** | 2.19 | 2.84 | 2.41 | 3.13 | 2.80 |
| | **OLG** | 0.28 | 0.67 | 2.30 | 2.29 | 0.28 |
| **South Africa** | **SSP** | 2.06 | 2.75 | 2.35 | 3.03 | 2.62 |
| | **OLG** | −0.90 | −0.27 | 3.15 | 3.72 | −0.90 |
| **USA** | **SSP** | 3.15 | 3.54 | 2.42 | 4.09 | 4.27 |
| | **OLG** | 3.90 | 4.42 | 6.98 | 8.83 | 3.91 |

## Appendix F. Additional Results

Comparing the probability of migration and migration flows across time in different SSP scenarios in Figure A9 shows that:

- In the SSP1, SSP2, and SSP5 scenarios, although low-skilled and high-skilled migration probabilities are in the same range, high-skilled migrants come from a faster growing population base;
- In the SSP3 and SSP4 scenarios, low-skilled migration probabilities are lower than high-skilled ones, while low-skilled migrants come from countries with larger populations.

As a result, high-skilled migration is more dominant in SSP1, SSP2, and SSP5, while low-skilled migration is larger in SSP3 and SSP4, highlighting the fact that *inequality effect* is more dominant among low-skilled migrants. Several empirical studies have shown that the number of high-skilled laborers is potentially larger than the number of low-skilled migrants. The reason however, is that high-skilled workers have access to greater financial resources compared to their low-skilled counterparts, which enables them to migrate in larger numbers [68,69].

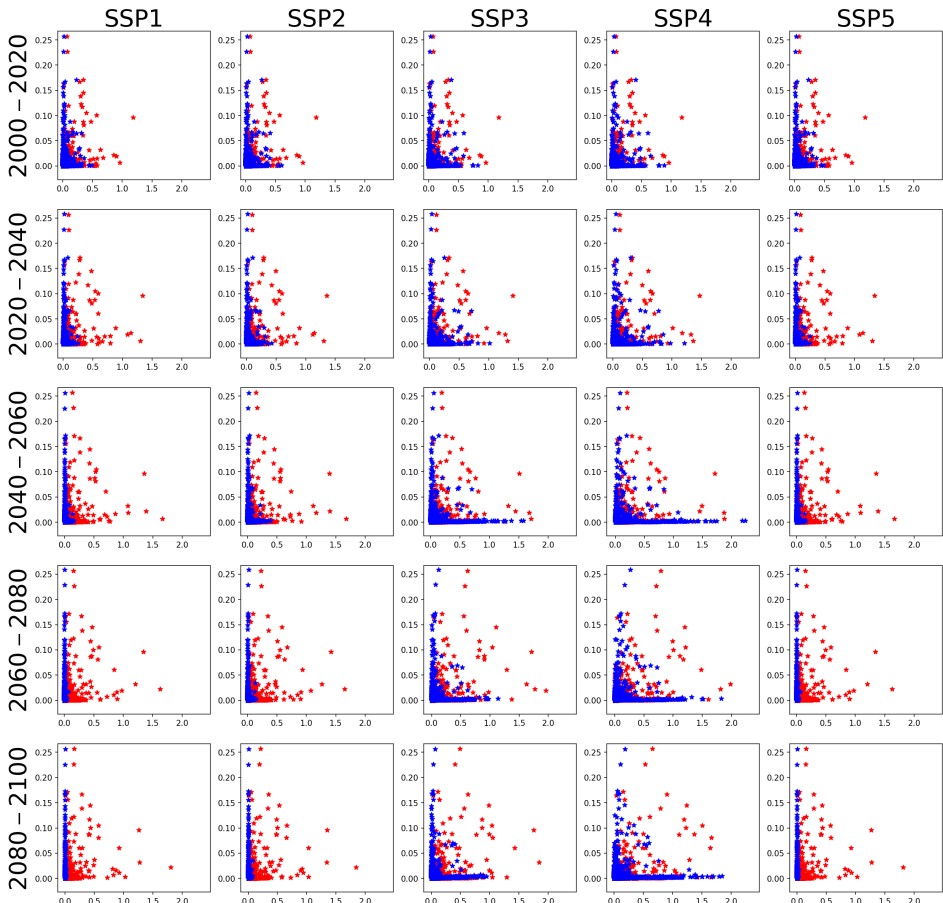

**Figure A9.** Evolution of bilateral migration flows in millions of people (*X*-axis) with migration probabilities (*Y*-axis) across time and SSPs. Migration flows between each pair of countries are represented by two dots, with high-skilled migration in red and low-skilled migration in blue.

Finally, the flow of bilateral migration can exert population stress in both sending and receiving countries. The ratio of migrants to the adult populations of sending and receiving countries under the five SSP scenarios is depicted in Figure A10 for the 2060–2080 period. Comparing historical stress levels from the 1980–2000 period with the future projections reveals that although historical destinations such as North America and Europe will continue to attract international immigrants, new regional hot-spots in Sub-Saharan Africa and Asia are emerging. Meanwhile, India and Brazil will experience higher rates of emigration compared to their historical levels. Table A6 in Appendix D.4 provides a comparison between the results of our model and the underlying migration assumptions for selected countries in each SSP scenario In general, the OLG model generates larger migration ratios than the underlying SSP assumptions. There are several explanations for such discrepancies. First, the population in the OLG model is endogenous, while the model follows the SSP assumptions about the growth rate of skill ratio. This leads to a sharp increase in the share of high-skilled labor over the next few decades, which in turn reduces the fertility in these countries. However, the SSP population growth assumptions are not necessarily held back by such a trade-off. Second, we did not include any migration policy changes in our baseline model. Therefore, in the absence of a detailed migration policy projection for each country and skill level, we believe our results can provide an estimated upper bound on migration flows.

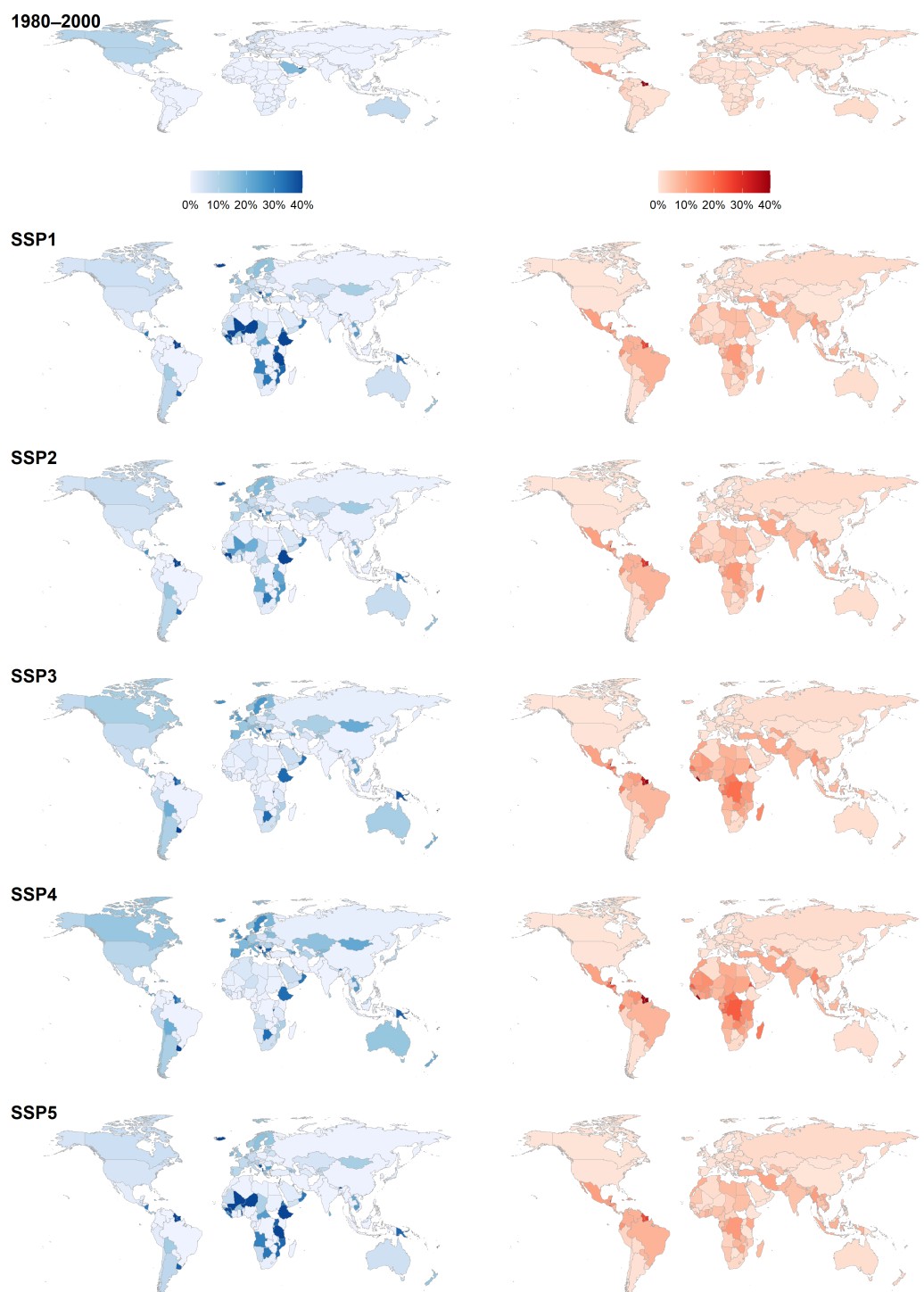

**Figure A10.** Ratio of migrants to adult population in receiving (blue) and sending (red) regions in the 1980–2000 and 2060–2080 periods across the SSPs.

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
