# Peer review of "International Migration Projections across Skill Levels in the Shared Socioeconomic Pathways"

_sustainability, doi:10.3390/su14084757_

Round 1

Reviewer 1 Report

Dear Authors,

Our minor recommendations are the following:

  • to include the limits of the study
  • to develop the Discussions section that has for now only some synthetic conclusions
  • to develop the research perspectives of the paper in the Conclusion section

Reviewer 2 Report

My criticisms of the manusscript "International Migration Projections across skill levels in the Shared Socioeconomic Pathways", are as follows:

- The theoretical foundations of the study are rather weak. There should be detailed theoretical explanations in the introduction.
- The contribution of the study should be explained in detail.
- The literature section is rather weak. It should be developed.
- The reasons for preferring the econometric method should be explained in detail.
- The conclusion part should be rewritten in accordance with the findings obtained.

Reviewer 3 Report

The topic of international migration is very important and current.

In the conclusion, the authors should add limitations and recommendations for future research.

Will this model be applicable to the analysis of the refugee situation observed, for example, as a result of the war in Ukraine?

Reviewer 4 Report

I would strongly recommend publishing this paper after some small changes

  1. the literature review section might be a little more detailed. Authors have taken under consideration wide range of factors determining migrations and their impact on different social groups, so it needed to be properly described in the theoretical section
  2. "Discussion" is connected with "Conclusions" and actually is only to conclude presented results. It definitely should be developed according to required paper structure

Round 2

Reviewer 2 Report

accept